# Q-Sched: Pushing the Boundaries of Few-Step Diffusion Models with Quantization-Aware Scheduling

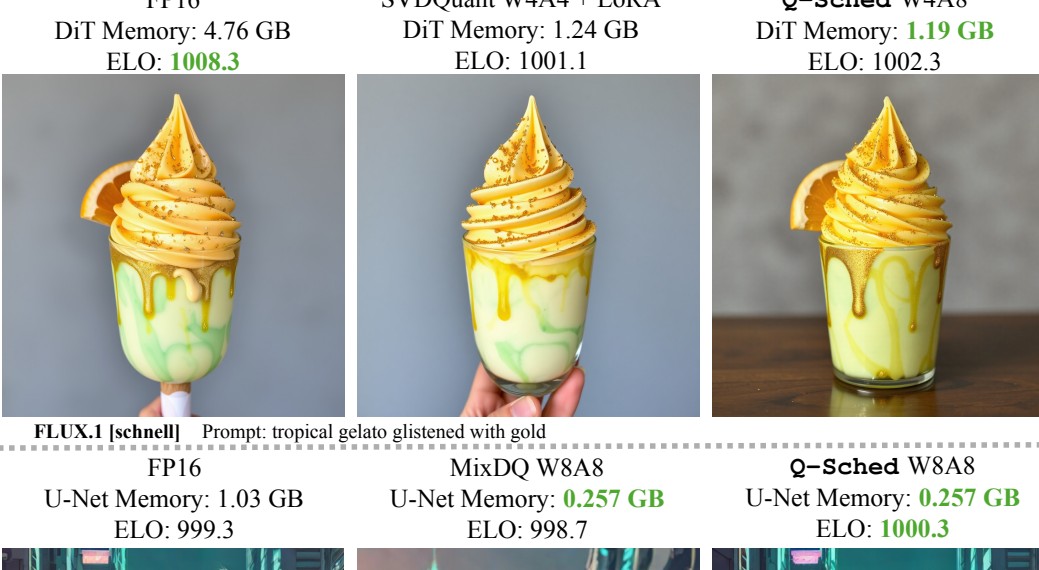

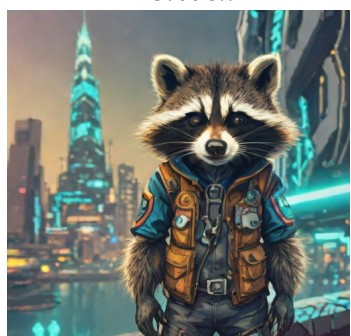 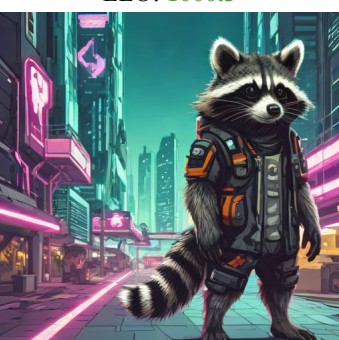

Figure 1: When large diffusion models are reduced to W8A8 or W4A8 for deployment, image fidelity drops. `Q-Sched` applies scheduler-level tuning, just two coefficients per step, to steer the sampler back to FP16-like quality, with no new checkpoints, no finetuning, and no extra FLOPs.

## ABSTRACT

Text-to-image diffusion models remain computationally intensive: generating a single image typically requires dozens of passes through large transformer backbones (*e.g.*, SDXL uses $\sim 50$ evaluations of a 2.6B-parameter model). Few-step variants reduce the step count to 2–8, but still rely on large, full-precision U-Net/DiT backbones, making inference impractical on resource-constrained platforms, both on-device (latency/energy) and in data centers with multi-instance GPU (MIG) style GPU partitioning (limited memory/throughput per slice). Existing post-training quantization (PTQ) methods are further hampered by dependence on full-precision calibration.

We introduce `Q-Sched`, a scheduler-level PTQ approach that adapts the diffusion sampler rather than the model weights. By adjusting the few-step sampling tra-

jectory with quantization-aware preconditioning coefficients, Q-Sched matches or surpasses full-precision quality while delivering a $4\times$ reduction in model size and preserving a single reusable checkpoint across bit-widths. To learn these coefficients, we propose a reference-free Joint Alignment–Quality (JAQ) loss, which combines text–image compatibility with an image-quality objective for fine-grained control; JAQ requires only a handful of calibration prompts and avoids any full-precision inference during calibration.

Empirically, `Q-Sched` yields substantial gains: a **15.5%** FID improvement over the FP16 4-step Latent Consistency Model and a **16.6%** improvement over the FP16 8-step Phased Consistency Model, demonstrating that quantization and few-step distillation are complementary for high-fidelity generation. A large-scale user study with **80,000+** annotations further validates these results on both FLUX.1[schnell] and SDXL-Turbo. Code will be released.

# 1 INTRODUCTION

Diffusion models have achieved state-of-the-art generative quality across vision (Amit et al., 2021; Baranchuk et al., 2021; Brempong et al., 2022; Ho et al., 2022; Meng et al., 2021; Yang et al., 2022a), language (Austin et al., 2021; Li et al., 2022b), multimodal modeling (Avrahami et al., 2022; Ramesh et al., 2022), and scientific domains (Anand & Achim, 2022; Cao et al., 2022). Yet systems such as Stable Diffusion XL (Podell et al., 2023; Meng et al., 2021) and CogVideoX (Yang et al., 2024) remain costly at inference time: denoising typically requires tens to hundreds of steps, each invoking a large U-Net or Diffusion transformer (DiT) (Peebles & Xie, 2023).

Practical deployment therefore hinges on two levers: (1) reducing the number of function evaluations (few-step sampling), and (2) lowering the cost per evaluation (compression via quantization (He et al., 2024; Guo et al., 2022), pruning (Fang et al., 2024), or distillation (Huang et al., 2024)). These levers are particularly important in two widely used settings. *On-device*, memory and compute budgets are tight, latency and energy constraints are strict, and privacy/offline use cases preclude server offloading (Zhao et al., 2024b). *In data centers with MIG partitioning*, a single GPU is sliced into multiple smaller instances to increase concurrency and predictability; each slice has limited memory/throughput, making model footprint and per-step cost decisive (Zhang et al., 2023; Li et al., 2022a). In both cases, few-step sampling and quantization are natural, complementary choices.

However, few-step acceleration is sensitive to the accuracy of the underlying probability-flow ordinary differential equation (ODE) or variance-preserving stochastic differential equation (SDE) that links the noise-estimation network to the final sample (Song et al., 2021). Quantization perturbs that network, inducing a mismatch that alters the ODE/SDE trajectory and amplifies artifacts, an effect that becomes more pronounced as the number of steps shrinks. Simply reusing full-precision schedulers on quantized backbones will inevitably induce quality degradation.

To bridge this gap, we introduce **Q-Sched**, a quantization-aware noise scheduler that adapts the few-step trajectory to the compressed model *without modifying any weights*. Q-Sched inserts lightweight coefficients $(\mathbf{c}^{\mathbf{x}}, \mathbf{c}^{\epsilon})$ into the scheduler (Figures 2a and 2b), correcting quantization-induced drift while keeping a single U-Net/DiT checkpoint reusable across FP16, W8A8, and W4A8 deployments. This design directly targets the constraints above: it preserves the latency benefits of few-step sampling, fits within on-device and MIG memory budgets, and avoids checkpoint sprawl in production.

Our contributions are summarized as follows:

1. In this work, we introduce **Q-Sched**, a quantization-aware scheduler that integrates seamlessly with few-step diffusion models. It achieves up to a **15.5% FID improvement** over a 4-step latent consistency model (LCM) (Luo et al., 2023) baseline and, as shown in Figure 1, can match or surpass full-precision arena scores *while simultaneously reducing model size* on SDXL-Turbo (4-Step) (Sauer et al., 2024) and FLUX.1[schnell] (Black Forest Labs, 2024).

2. Q-Sched's novel **preconditioning coefficients** enable quantized models to deliberately deviate from potentially overfit few-step baselines (Figure 2a), alleviating oversmoothing

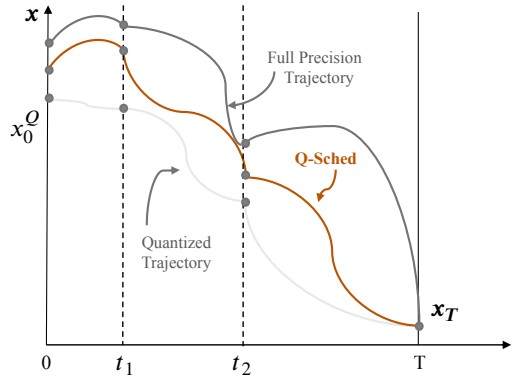
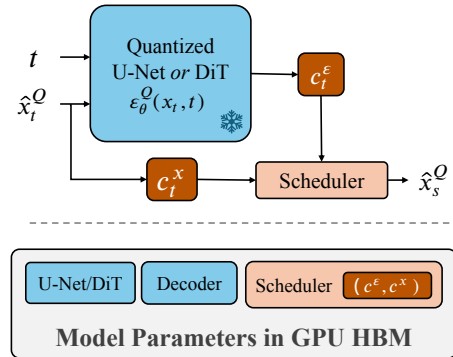

(a) Quantization shifts the diffusion sampling trajectory, reducing fidelity. Q-Sched corrects this drift by adapting the scheduler.

(b) At timestep $t \to s$, Q-Sched applies lightweight coefficients $(c_t^x, c_t^\epsilon)$ within the scheduler, enabling deployment of quantized models from a single U-Net/DiT checkpoint.

Figure 2: Q-Sched directly optimizes the few-step diffusion scheduler (see Figures 2a and 2b), addressing quantization-induced trajectory drift without modifying model weights. Unlike prior approaches that alter the transformer or U-Net backbone through retraining or post-training adjustments, Q-Sched leaves weights fixed, allowing seamless reuse of one pretrained checkpoint across FP16, W8A8, and W4A8 deployments. This simplifies model management and reduces storage overhead while maintaining high image fidelity.

    and texture artifacts from distillation and quantization while improving the balance between fidelity and artifact severity.

3. To optimize these coefficients, we propose the **Joint Alignment–Quality (JAQ) loss** which balances perceptual fidelity with text–image alignment. Being reference-free, JAQ also enables precise control over visual properties (*e.g.*, texture, detail, saturation) without requiring access to a full-precision model.

4. We establish a **theoretical existence guarantee** (Theorem 1), proving that Q-Sched coefficients always exist which reduce expected sampling error relative to the original quantized scheduler. This provides a principled explanation for Q-Sched 's systematic improvements.

5. Finally, a large-scale **human preference study** with over 80,000 annotations demonstrates that Q-Sched outperforms MixDQ (Zhao et al., 2024a) on SDXL-Turbo and SVDQuant (Li et al., 2025) on FLUX.1[schnell] in terms of perceived image quality.

As illustrated in Figure 1, Q-Sched attains the highest ELO rating in pairwise image-quality comparisons among evaluated methods. Furthermore, Figure 3 shows that Q-Sched is Pareto-optimal with respect to both ELO and model size, underscoring its ability to balance perceptual quality and efficiency more effectively than competing approaches.

## 2 BACKGROUND AND RELATED WORK

Diffusion models generate samples by denoising corrupted data across a trajectory of timesteps $t \in [0, T]$, where $T$ is typically large ($\geq 25$). Each step applies a denoising network $\mathcal{E}_\theta$, conditioned on both $t$ and its noisy input $x_t$. While this iterative scheme yields high-fidelity samples, invoking a large U-Net or DiT backbone at every step makes inference prohibitively slow in deployment.

Recent few-step models highlight this bottleneck. **SDXL-Turbo** leverages Adversarial Diffusion Distillation (ADD), combining score distillation with an adversarial loss, to reduce sampling to just 1–4 steps, enabling real-time generation on commodity GPUs (Sauer et al., 2024). **FLUX.1[schnell]** introduces a 12B-parameter rectified-flow transformer with open weights, optimized for 1-4 step inference, making it attractive for latency-constrained serving (Black Forest Labs, 2024). Most recently, **FLUX.1[kontext]** extends the family beyond text-to-image toward *in-context* generation and

editing, accepting text and images jointly and unifying both tasks in a flow-matching framework (Labs et al., 2025). These advances exemplify the field's shift toward *deployment-ready* diffusion systems that meet strict latency and memory budgets.

**Few-step diffusion and distillation.** Few-step methods compress the teacher's long trajectory into a handful of evaluations, preserving most of the fidelity at a fraction of the cost. Distillation is the primary approach: early demonstrations distilled long-run teachers into 1–8 step students, such as Instaflow (Liu et al., 2023), rectified-flow straightening (Liu et al., 2022), and adversarially guided ADD (Sauer et al., 2024). Consistency Models (CMs) (Song et al., 2023) frame generation as a self-consistency mapping from any noisy state to the clean sample, yielding efficient few-step samplers. Variants include Latent Consistency Models (LCMs) (Luo et al., 2023) with Stable Diffusion (Rombach et al., 2022) backbones, Trajectory Consistency Distillation (TCD) (Zheng et al., 2024) with trajectory-aware schedules, and Phased Consistency Models (PCMs) (Wang et al., 2024) with improved guidance and stability. Across these designs, the *scheduler* plays a critical role in determining quality in the few-step regime. The update rule for few-step diffusion models using quantized backbone $\mathcal{E}_\theta^Q$ is:

$$x_s = \Phi(t, x_t, \mathcal{E}_\theta^Q), \tag{1}$$

where $x_s$ denotes the intermediate sample at timestep $s \in [0, t]$ and $\Phi(\cdot)$ is a few-step scheduler. In Section 3, we illustrate our approach using the TCD scheduler (Zheng et al., 2024) as a running example. However, Q-Sched is fully general and can be applied on top of any few-step scheduler that fits the abstraction in Equation (1).

**Quantization for diffusion models.** Post-training quantization (PTQ) has largely targeted $\mathcal{E}_\theta$ and its activations across timesteps. Timestep-aware calibration approaches (PTQ4DM (Shang et al., 2022), ADP-DM (Wang et al., 2023a), Q-Diffusion (Li et al., 2023)), dynamic schemes such as TDQ (So et al., 2024), and error-compensation methods (Q-DM (Li et al., 2024c)) all operate by modifying weights or activations and require full-precision calibration. MixDQ (Zhao et al., 2024a) extends to few-step models with a mixed-precision allocation strategy guided by beggining-of-sentence(BOS)-aware quantization and layer sensitivity analysis. SVDQuant (Li et al., 2025) targets 4-bit weights and activations by absorbing outliers into a high-precision low-rank branch via SVD, shifting variance from activations into weights before fusing the branch back into low-bit kernels.

We posit that in the few-step setting, quantization bias additionally manifests as a *scheduler mismatch*: a fixed full-precision schedule can systematically over- or under-correct, amplifying artifacts. One method that avoids modifying network weights is PTQD (He et al., 2024), which models the quantization-induced shift as an affine perturbation of the full-precision denoiser, $\mathcal{E}_\theta^Q(x_t, t) = (1 + \gamma)\mathcal{E}_\theta + \delta$, and compensates it via variance scaling and a bias term applied directly to the sampler update on $x_t$. In practice, $\gamma$ is estimated via standard-deviation matching while $\delta$ is treated as uncorrelated Gaussian noise. We adapt PTQD-style bias correction, originally developed for *un-distilled* diffusion models, into TCD (see Section H) and generalize the principle to other few-step samplers as a baseline for our approach.

Q-Sched reframes quantized few-step generation as scheduler adaptation. It learns quantization-aware preconditioning coefficients to correct trajectory drift with negligible overhead, while leaving the backbone frozen. The approach integrates seamlessly with few-step schedulers, needs only lightweight calibration, and preserves a single checkpoint across FP16, W8A8, and W4A8. Unlike prior PTQ methods that adjust weights or activations, Q-Sched adapts the scheduler itself, complementing existing PTQ and distillation techniques to recover full-precision quality at reduced footprints while retaining the latency benefits of few-step sampling.

Q-Sched differs fundamentally from prior bias and variance scaling methods like PTQD (He et al., 2024) by learning its correction coefficients end-to-end using final image quality, rather than relying on Gaussian assumptions or intermediate denoising states. By introducing a second coefficient on $x_t$ and separating accumulated state error from current-step noise error, Q-Sched gains the flexibility to correct both sources of distortion independently—crucial for few-step distilled models where intermediate distributions are no longer Gaussian. Overall, Q-Sched removes dependency on full-precision activations, relaxes prior assumptions, and directly optimizes for the final output rather than intermediate signals.

## 3 Quantization-Aware Scheduling

To prepare the TCD scheduler for optimization with `Q-Sched`, let us consider sampling with a quantized network. TCD's Strategic Stochastic Sampling (SSS) (Zheng et al., 2024) using a quantized network $\mathcal{E}_\theta^Q(x_t, t)$ is given by:

$$\mathbf{x_s} = \frac{\alpha_s}{\alpha_{s'}} \left( \alpha_{s'} \frac{\mathbf{x_t} - \sigma_t \mathcal{E}_\theta^Q(x_t, t)}{\alpha_t} + \sigma_{s'} \mathcal{E}_\theta^Q(x_t, t) \right) + \eta \mathbf{z} \tag{2}$$

where the noise schedule is given by $\sigma, \alpha$ and the sampler injects stochastic noise sampled from a distribution $\mathbf{z} \sim N(0, I)$. The sampler relies on an intermediary timestep, $s' \in [s, t]$, where stochastic noise is added. The degree of randomness is controlled by the stochastic control parameter $\eta$:

$$\eta = \sqrt{1 - \frac{\alpha_s^2}{\alpha_{s'}^2}} \quad . \tag{3}$$

which can be adjusted at sampling time to vary image randomness. The TCD sampler in Equation (2), used in Phased Consistency Models, is a state-of-the-art few-step diffusion method that depends on two inputs from the previous step—$x_t$ and $\mathcal{E}_\theta^Q(x_t, t)$—which are central to applying `Q-Sched`.

**`Q-Sched`: A Learnable Schedule Pre-Conditioner**   We introduce `Q-Sched`, a lightweight post-training method that adapts the noise schedule of few-step diffusion models using two learnable scalar preconditioning coefficients, $c_t^x$ and $c_t^\epsilon$, applied respectively to $x_t$ and $\mathcal{E}_\theta^Q(x_t, t)$ at time $t$. As illustrated in Figure 2b, `Q-Sched` operates independently of the model backbone (U-Net or transformer), making it broadly compatible with any few-step scheduler resembling TCD.

Under `Q-Sched`, the TCD sampling update becomes:

$$\mathbf{x_s} = \frac{\alpha_s}{\alpha_{s'}} \left( \alpha_{s'} \frac{c_t^x \mathbf{x_t} - \sigma_t c_t^\epsilon \mathcal{E}_\theta^Q(x_t, t)}{\alpha_t} + \sigma_{s'} c_t^\epsilon \mathcal{E}_\theta^Q(x_t, t) \right) + \sqrt{1 - \frac{\alpha_s^2}{\alpha_{s'}^2}} \mathbf{z}. \tag{4}$$

In Equation (4), we explicitly show that `Q-Sched` interacts with the update rule of common few-step scheduler. Because the update rule is affine in the preconditioning coefficients, $(\mathbf{c^x}, \mathbf{c^\epsilon}) := (c_t^x, c_t^\epsilon)_{t=0}^T$, they can be fused into the existing TCD coefficients without modifying the computational graph or adding inference cost. In the TCD formulation, the estimate at the proxy timestep $s'$ is not obtained from a separate model evaluation but is derived directly from the prediction at timestep $t$. As a result, the noise term associated with $s'$ inherits the same coefficient $c_t^\epsilon$.

To learn $(\mathbf{c^x}, \mathbf{c^\epsilon})$, we perform hyperparameter search as outlined in Algorithm 1. While we describe the algorithm in terms of a generic optimizer, in practice, we find grid search is sufficient, as each model involves only two coefficients per timestep across 2–8 timesteps. In the grid search setting, `opt.step` simply advances to the next point in the predefined search grid. Even small adjustments to these coefficients yield noticeably crisper images with fewer quantization artifacts. More details about `Q-Sched`'s search is in Section M.

A natural question arises: **why are two coefficients sufficient to improve image quality?** It turns out that the reconstruction error between full precision and quantized images (at timestep $t = 0$), denoted by $\Delta x_0$, can be strictly improved using scheduler coefficients:

**Theorem 1** (Strict Existence Guarantees). *There exists Q-Sched coefficients* $(\mathbf{c^x}, \mathbf{c^\epsilon}) \neq 0$ *such that* $E[||\Delta \tilde{x}_0||] < E[||\Delta x_0||]$.

As shown in Appendix I, $\Delta x_0$ is a linear combination of per-step denoising errors $\Delta E_\theta(t)$ with coefficients $k_t, m_t$. Since the error is homogeneous in these terms, rescaling via $\tilde{k}_t = c_t^x k_t$ and $\tilde{m}_t = c_t^\epsilon m_t$ strictly reduces the expected error over naïve quantization. Thus, re-weighting the sampler, without modifying network weights, guarantees a reduction in error with respect to the full precision images. Next, we will discuss our new reference-free loss function, JAQ, and its advantages over existing image assessment tools.

---

**Algorithm 1** Search for `Q-Sched` Coefficients

**Input:** search range $[c_{min}, c_{max}]$, search points $n$, number of diffusion steps $\omega$
loss function JAQ, calibration set $\mathcal{C}$, search optimizer `opt`

1: Initialize $S^* \leftarrow \infty$
2: ▷ initialize each parameter in uniformly distributed range $(c_{min}, c_{max})$
3: $(c^x_{start}, c^x_{end}, c^\epsilon_{start}, c^\epsilon_{end}) \leftarrow$ `opt.init`$(c_{min}, c_{max})$
4: **for** $i \in [0, n]$ **do**
5: $\quad \mathbf{c^x} \leftarrow$ `linspace`$(c^x_{start}, c^x_{end}, \omega)$
6: $\quad \mathbf{c^\epsilon} \leftarrow$ `linspace`$(c^\epsilon_{start}, c^\epsilon_{end}, \omega)$
7: $\quad S \leftarrow [\,]$
8: $\quad$ **for** $x \in \mathcal{C}$ **do**
9: $\quad\quad S_x \leftarrow$ JAQ$(x; \mathbf{c^x}, \mathbf{c^\epsilon})$
10: $\quad\quad S = S \cup S_x$
11: $\quad$ **end for**
12: $\quad$ ▷ $\bar{S}$ is the arithmetic mean of S
13: $\quad$ **if** $\bar{S} < S^*$ **then**
14: $\quad\quad S^* \leftarrow \bar{S}$ , $\mathbf{c^x_\star} \leftarrow \mathbf{c^x}$, $\quad \mathbf{c^\epsilon_\star} \leftarrow \mathbf{c^\epsilon}$
15: $\quad$ **end if**
16: $\quad (c^x_{start}, c^x_{end}, c^\epsilon_{start}, c^\epsilon_{end}) \leftarrow$ `opt.step`$(\bar{S})$
17: **end for**
18: **return** $\mathbf{c^x_\star}, \mathbf{c^\epsilon_\star}$

---

**JAQ: A Joint Alignment Quality Loss Function**   Because full precision intermediate states are unstable targets due to quantization-induced structural and semantic drift, optimizing directly for downstream image quality provides a far more reliable objective than attempting to match the full precision trajectory. Reference-free metrics such as CLIPScore (Hessel et al., 2021) have become essential for quick evaluation of text-to-image generation models and unlike FID (Heusel et al., 2017), SSIM (Wang et al., 2004), and other *comparative* metrics, reference-free metrics do not rely on a ground truth reference image and therefore are very useful in this setting. When quantizing these generative models, the resultant images, $\hat{x}_0^Q$, are generated by an altered sampling trajectory as evidenced in Figure 8, where $\hat{x}_0^Q$ is a different, sometimes cleaner image than a those derived from the full precision backbone. In short, the quantized model's sampling trajectory coarsely follows the full precision model yet generates sufficient differences that reference-based metrics do not capture the image's detail.

Our Joint Alignment Quality loss combines a text-to-image compatibility score with a pure image quality score to achieve better results than simply optimizing with respect to metrics such as CLIP-Score or CLIP-IQA (Wang et al., 2023b) independently. We design the JAQ loss so that it can better differentiate between images that are highly similar to one another, whereas standard image quality metrics are designed to rank images that come from a much larger distribution. Given a text-to-image compatibility metric, `TC`$(x)$, and a pure image quality metric, `IQ`$(x)$, JAQ combines them as follows:

$$\text{JAQ}(x) = \text{TC}(x) + k \cdot \text{IQ}(x) \tag{5}$$

Optimizing solely for text–image compatibility (e.g., CLIPScore) sacrifices visual detail and fails to capture quantization artifacts (Figures 7 and 9). Conversely, relying only on image quality can generate extraneous details. JAQ balances these objectives through a linear combination, with $k$ controlling the tradeoff between prompt fidelity and image detail.

**Applying `Q-Sched` to Full Precision Models?**   While it is theoretically possible that applying Q-Sched to a non-quantized model could yield some improvement, this is not the setting the method is designed for. A full precision model is best optimized directly through its original training objectives such as distillation and fine-tuning rather than through a post-training scheduler adjustment. Q-Sched specifically targets post training degradation introduced by quantization and is intended to correct the resulting drift in the sampling trajectory.

Table 1: Comparison of different schedulers on Phased Consistency Models and Latent Consistency Models using a Stable Diffusion v1-5 backbone. The original schedule is TCD (Zheng et al., 2024) for Phased Consistency Models and the Multi-step Consistency Sampling (Luo et al., 2023) for Latent Consistency Models. The FID and CLIPScore are calculated with respect to the COCO-30k dataset. NFEs stands for *number of function evaluations* referring to the number of passes through the network $\mathcal{E}_\theta^Q(x_t, t)$. We report latency in milliseconds on an RTX A6000 GPU.

| NFEs | Precision | Schedule | Latency (ms) | PCMs | | LCMs | |
|---|---|---|---|---|---|---|---|
| | | | | FID | CLIPScore | FID | CLIPScore |
| 2 | FP16 | Original | 148 | 24.17 | 25.489 | 38.74 | 25.155 |
| | W4A8 | Original | 136 | 28.70 | 25.343 | 40.93 | 24.886 |
| | W4A8 | Q-Sched | 137 | 23.33 | 25.265 | 37.59 | 24.919 |
| | W4A8 | Q-Sched | 136 | **22.24** | **25.543** | **32.50** | **25.152** |
| 4 | FP16 | Original | 193 | 23.29 | 25.482 | 31.94 | **25.969** |
| | W4A8 | Original | 172 | 23.08 | 25.557 | 38.41 | 25.456 |
| | W4A8 | PTQD | 172 | 19.42 | 25.639 | 39.72 | 24.678 |
| | W4A8 | Q-Sched | 172 | **17.39** | **25.715** | 26.98 | 25.336 |
| 8 | FP16 | Original | 286 | 20.15 | **25.714** | 27.34 | **26.052** |
| | W4A8 | Original | 245 | 18.48 | 25.664 | 27.55 | 25.397 |
| | W4A8 | PTQD | 246 | **15.85** | 25.770 | 28.06 | 25.241 |
| | W4A8 | Q-Sched | 245 | 16.83 | 25.698 | **25.82** | 25.214 |

## 4 EXPERIMENTS

**Experimental Setup** We apply Q-Sched across diverse few-step diffusion models, including U-Net (Ronneberger et al., 2015) and DiT (Peebles & Xie, 2023) backbones, and across different distillation strategies: consistency-based (LCM (Luo et al., 2023), PCM (Wang et al., 2024)) and flow-matching approaches (SDXL-Turbo (Sauer et al., 2024) and FLUX.1[schnell] (Black Forest Labs, 2024)). We quantize models in both 4-bit weights, 8-bit activations (W4A8) and 8-bit weights, 8-bit activations (W8A8). Only the U-Net or DiT backbone is quantized, as it dominates model size (see Table 5).

Latency is measured on an Nvidia RTX A6000 GPU with Ampere compute architecture. Using BitsandBytes bit (2025), we quantize each model to 4-bit weights, 8-bit activations and average latency over 10 runs with a 3-run warmup phase.

LCM and PCM are tested at 2, 4, and 8 steps on COCO-30k (Lin et al., 2014), using FID (vs. real), CLIPScore (prompt alignment), and FID-SD (vs. Stable Diffusion). FLUX.1 and SDXL-Turbo are evaluated on the SVDQuant (Li et al., 2025; 2024b) subset of MJHQ-30k (5,000 high-quality Midjourney prompts in 10 categories), using FID and human preference studies to capture perceptual quality.

We employ two variants of the Joint Alignment Quality (JAQ) loss: one derived from CLIP-based metrics and another from human preference scores. In the CLIP-based variant, we set TC(x) = CLIPScore(x) and IQ(x) = CLIP-IQA(x). For SDXL-Turbo and FLUX.1, we instead adopt a preference-based variant, with TC(x) = AQ-MAP(x) and IQ(x) = HPSV2(x). Here, AQ-MAP (Li et al., 2024a) provides a spatial alignment score, while HPSV2 (Wu et al., 2023) is fine-tuned on real human judgments. In both cases, we fix $k = 2$.

**Results: Latent and Phased Consistency Models** In Table 1, we evaluate three schedulers across two consistency model families and show that Q-Sched learns a new few-step trajectory that mitigates artifacts and can even surpass both FP16 and W4A8 in detail. It achieves strong FID scores and outperforms PTQD in 4/6 consistency variants on Stable Diffusion v1-5, while using only a fraction of the calibration set. We compare with PTQD (He et al., 2024), the only other quantization-aware scheduler for few-step diffusion. Unlike PTQD, which relies on a 1,024-image full-precision calibration set, Q-Sched requires only 20 representative sDCI prompts (Li et al., 2025), reused across evaluations. Calibration overfitting is a common issue in PTQ methods, but

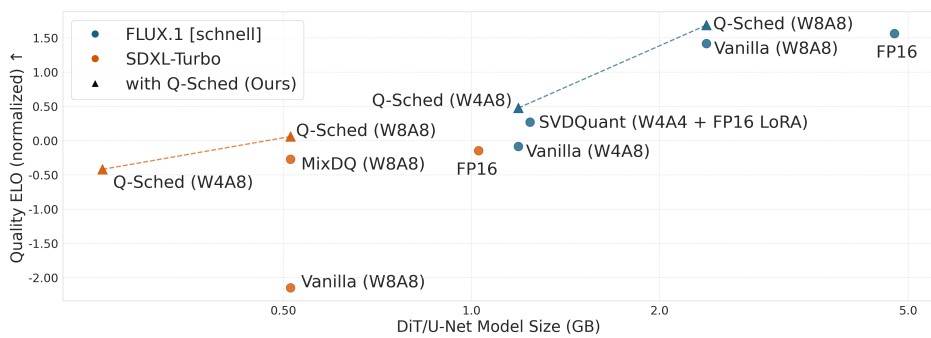

Figure 3: ELO Score *vs.* Model Size for various quantization methods on FLUX.1[schnell] (Black Forest Labs, 2024) and SDXL-Turbo (Sauer et al., 2024).

we mitigate it by using a highly descriptive long-form sDCI calibration set and optimizing only a small number of coefficients. This helps the quantized model learn to handle complex scenes, which generalizes well to simpler prompts. Our results show that this strategy yields strong performance on MJHQ and COCO-30k, two distinct downstream datasets. Unlike PTQD, which requires full-precision references, Q-Sched operates with just twenty prompts and can exceed a full precision few-step model by **16.1%, 15.5%, and 5.6%** at 2, 4, and 8 steps, respectively. This highlights that quantization and few-step distillation act as complementary compression strategies.

Furthermore, we would like to emphasize that Q-Sched and PTQD do not add any additional latency overhead to the sampler, since their coefficients can be fused to the existing schedule at inference time.

| Scheduler | Precision | FID | FID-SD | CLIPScore |
|---|---|---|---|---|
| TCD | FP16 | 18.65 | 10.45 | 26.531 |
| TCD | W4A8 | 22.70 | 12.51 | 26.241 |
| PTQD | W4A8 | 161.96 | 176.29 | 25.910 |
| Q-Sched | W4A8 | **18.89** | **12.17** | **26.513** |

(a) Comparison on a 2-step Phased Consistency model using the Stable Diffusion XL backbone. FID-SD is computed relative to images generated by Stable Diffusion XL using corresponding COCO-30k prompts.

| Method | Precision | FID |
|---|---|---|
| - | FP16 | 25.48 |
| Naive | W4A8 | 25.75 |
| MixDQ | W4A8 | 25.36 |
| Q-Sched | W4A8 | **21.41** |
| Naive | W8A8 | 25.49 |
| MixDQ | W8A8 | **25.16** |
| Q-Sched | W8A8 | 26.34 |

(b) Quantized model comparison on SDXL-Turbo under varying bitwidths. FID is computed on the MJHQ dataset.

Table 2: Quantitative evaluation of Large-scale few-step diffusion models with a Stable Diffusion XL backbone. W4A8 and W8A8 are a $4\times$ and $8\times$ model size reduction in comparison to FP16, yet our method improves over baseline. As FID, FID-SD, and CLIPScore may exhibit reduced reliability at large model scales, we complement these metrics with user preference studies in Figure 3.

In Table 2a, we evaluate a large-scale 2-step Phased Consistency Model on the Stable Diffusion XL backbone. Q-Sched incurs only a 1.2% FID drop in W4A8, showing that quantization-aware preconditioning preserves quality even under aggressive compression. By contrast, PTQD degrades sharply, as its Gaussian noise assumption breaks down in few-step diffusion—particularly for large models where each step approximates an ODE segment rather than a Gaussian denoising step.

**Results: SDXL-Turbo and FLUX.1[schnell]**  In Table 2b, we compare quantization strategies on SDXL-Turbo (4-step inference) using the FID metric on the MJHQ dataset, evaluating two bitwidth settings: W4A8 and W8A8. Under W4A8, Q-Sched achieves a FID of 21.41 with a standard deviation (std) of 0.15, significantly outperforming MixDQ (Zhao et al., 2024a) (25.36, std 0.17) and Naive (25.75, std 0.28), demonstrating strong robustness to aggressive quantization. Q-Sched

Table 3: Comparison across image quality metrics. "MixDQ" refers to the W8A8 MixDQ (Zhao et al., 2024a) variant and "SVDQ" refers to LoRA-based W4A4 SVDQuant Li et al. (2025).

| | SDXL-Turbo (4-Step) | | | | FLUX.1 [schnell] | | | |
|---|---|---|---|---|---|---|---|---|
| | FP16 | W8A8 | MixDQ | Q-Sched | FP16 | W4A8 | SVDQ | W4A8 Q-Sched |
| CLIP Score ↑ | **25.62** | **25.62** | 25.38 | 25.36 | **25.61** | 25.17 | 25.52 | 25.27 |
| CLIP IQA ↑ | 0.725 | 0.727 | 0.727 | **0.731** | **0.716** | 0.712 | 0.714 | 0.707 |
| HPV2 ↑ | 0.276 | 0.276 | 0.275 | **0.278** | **0.275** | 0.274 | **0.275** | 0.272 |
| AQ-MAP ↑ | 0.693 | 0.694 | 0.693 | **0.696** | **0.700** | **0.700** | 0.697 | **0.700** |
| Pick Score ↑ | 18.48 | 18.49 | 18.48 | **18.51** | 18.43 | 18.42 | 18.40 | **18.46** |
| MANIQA ↑ | 0.508 | 0.513 | 0.502 | **0.511** | **0.528** | 0.500 | 0.514 | 0.506 |
| JAQ (ours) ↑ | 1.663 | 1.665 | 1.659 | **1.669** | **1.676** | 1.675 | 1.669 | 1.673 |

learns its own sampling trajectory—often improving image quality despite FID fluctuations—and across diverse metrics it remains competitive with, and sometimes better than, both state-of-the-art quantization methods and full-precision models. However, at W8A8, `Q-Sched` shows a higher FID (26.34) than both MixDQ (25.16) and Naive (25.49), suggesting that its advantages are most pronounced in lower-bit regimes, where other methods degrade more severely.

In Figure 3, we present user preference results for `Q-Sched` applied to both SDXL-Turbo and FLUX.1 [schnell], showing that it outperforms MixDQ (Zhao et al., 2024a) and SVDQuant (Li et al., 2025), respectively, at similar model sizes (see Section B for details). We compute an ELO rating, a relative quality ranking inspired by chess scoring, by aggregating all pairwise 1v1 image comparisons across models, where a higher score reflects consistent user preference.

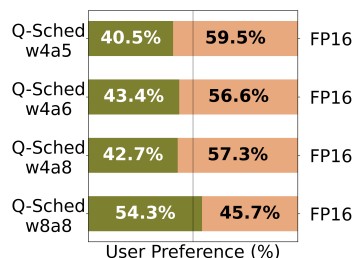

Figure 4: Comparing `Q-Sched` across various bit-widths.

In Figure 4, we compare `Q-Sched` across bit-widths using a user study. W4A4 proved too aggressive, but W4A5 and W4A6 produced images comparable to full precision. 1v1 comparisons with full-precision FLUX.1 (Black Forest Labs, 2024) follow the protocol in Appendix B.

**Comparison with Image Quality Metrics** We also evaluate `Q-Sched` on FLUX.1[schnell] and SDXL-Turbo using human preference metrics, showing that JAQ effectively captures image fidelity. Beyond the metrics we've already mentioned, we compare with PickScore (Kirstain et al., 2023) which predicts human preferences from large-scale image–text comparisons, and MANIQA (Yang et al., 2022b) which uses multi-dimensional attention to assess perceptual quality without references. As seen in Table 3, JAQ aligns closely with established metrics while uniquely balancing fine-grained details often degraded by quantization.

**Ablation on Pre-Conditioning Coefficients and Loss Function Choice** We ablate the choice of pre-conditioning coefficients in the Phased Consistency Model by comparing performance when optimizing only the model-side coefficient $\mathbf{c}^\epsilon$, the sample-side coefficient $\mathbf{c}^\mathbf{x}$, or both jointly. As shown in Figure 5b, jointly optimizing both $\mathbf{c}^\epsilon$ and $\mathbf{c}^\mathbf{x}$ consistently yields the best results across all three metrics: PickScore, HPSv2, and JAQ Loss. These findings highlight the importance of treating both denoising and reconstruction terms as tunable components rather than fixing one a priori. All metrics are averaged over 1024 images generated with the SDXL backbone.

**How Do We Choose $k$ For The JAQ Loss?** We optimize the `Q-Sched` preconditioners using the JAQ loss, which balances image quality and text-image consistency via a tradeoff hyperparameter, $k$. As shown in Figure 5a, small $k$ values can lead to color distortion, while larger values (*e.g.*, $k = 5$) cause outputs to drift from the true data distribution. In such cases, the JAQ loss behaves similarly to CLIP-IQA-Q, which lacks sensitivity to concept alignment. We find that a hand-tuned value of $k$ is sufficient for producing a high-quality noise schedule, and the final results are not highly sensitive to its exact choice. Throughout our experiments, we use $k = 2$.

|  | $c^\epsilon$ | $c^x$ | $(c^\epsilon, c^x)$ |
|---|---|---|---|
| PickScore | 21.83 | 22.25 | **22.30** |
| HPSV2 | **0.288** | 0.262 | **0.288** |
| JAQ Loss | 3.367 | 3.383 | **3.392** |

(b) Ablation on choice of pre-conditioning coefficients. We find that optimizing both model and sample coefficients jointly yields optimal image quality. Image quality metrics are averaged over 1024 images generated from the Phased Consistency Models with the SDXL backbone.

(a) Choice of $k$ for the JAQ loss. $k$ balances the contribution of $\texttt{TC}(x)$ *vs.* $\texttt{IQ}(x)$. Prompt: "a car and a bus on a french highway".

Figure 5: Ablation studies on various design choices for $\texttt{Q-Sched}$.

## 5 CONCLUSION

Few-step diffusion models dramatically reduce inference cost by distilling large generative models, such as Stable Diffusion XL, into versions requiring only 2–8 denoising steps, achieving a 5–25× speedup. However, these models typically reduce runtime without addressing model size. Our method, $\texttt{Q-Sched}$, pushes this efficiency frontier further by introducing quantization into the few-step regime. Through noise-aware preconditioning coefficients, $\texttt{Q-Sched}$ enables effective quantization with minimal performance loss. We report **8.0% and 16.1% FID improvements** over full-precision baselines for PCMs and LCMs, respectively. A user preference study also shows that **$\texttt{Q-Sched}$ outperforms existing quantization methods on FLUX.1[schnell] and SDXL-Turbo in perceived image quality**. These results demonstrate that quantization and few-step distillation are complementary, enabling substantial efficiency gains without compromising generation quality.

## 6 ETHICS STATEMENT

Model compression broadens the accessibility of AI by enabling large foundation models to run on resource-constrained GPUs. The potential societal consequences of our work are similar to those of prior approaches, as both quantization and few-step diffusion serve as compression methods for text-to-image generative models. Such models can produce synthetic images that may mislead, misrepresent, or cause social harm. We conduct a user preference study on a crowdsourcing platform in which participants worldwide are shown generated content, which, like all synthetic media, carries inherent potential for misuse and harm.

## 7 LLM USAGE

We made use of large language models (LLMs) to assist in the preparation of this manuscript. LLMs were employed for language polishing, formatting support (e.g., LaTeX macros, algorithm pseudocode, figure/table captions), and iterative feedback on clarity and conciseness of explanations.

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
