## A  Q-Sched Images

In Figure 6, we provide images comparing SVDQuant with Q-Sched on a W4A4 quantized FLUX.1[schnell].

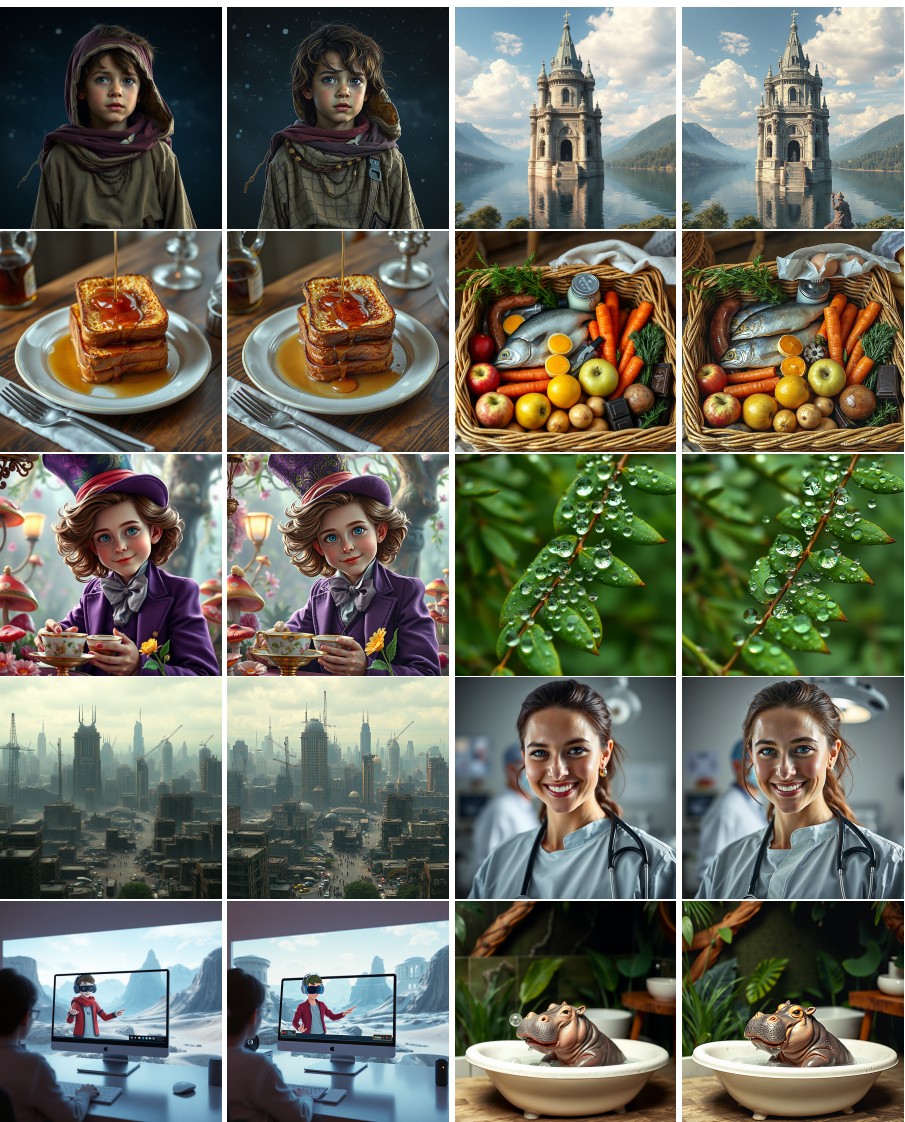

Figure 6: Side by side visual comparison of SVDQuant results and Q Sched corrected outputs for 10 MJHQ prompts. Each pair shows the SVDQuant output on the left and the corresponding Q Sched output on the right.

## B  Details on User Preference Assessment

We design our evaluation setup following the user preference study methodology from SDXL-Turbo (Sauer et al., 2024), with several improvements. For each model pair in this study, we perform 1-vs-1 comparisons based on shared prompts. Human responses, collected via Rapidata (Rapidata, 2025), come from evaluators who are presented with two images, each generated by a different model for the same prompt, and are asked: "Which image is of higher quality and more aesthetically pleasing?"

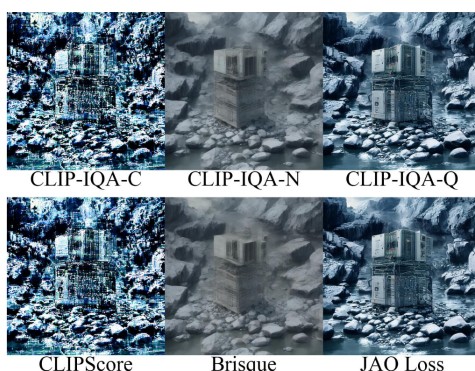
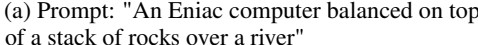
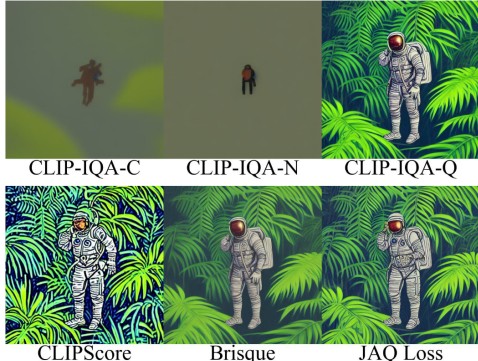

| CLIP-IQA-C | CLIP-IQA-N | CLIP-IQA-Q | | CLIP-IQA-C | CLIP-IQA-N | CLIP-IQA-Q |

| CLIPScore | Brisque | JAQ Loss | | CLIPScore | Brisque | JAQ Loss |

(a) Prompt: "An Eniac computer balanced on top of a stack of rocks over a river"

(b) Prompt: "Astronaut in a jungle, cold color palette, muted colors, detailed, 8k"

Figure 7: Optimizing `Q-Sched` with various reference-less image quality metrics. Our loss function, JAQ, is a linear combination of CLIPScore and CLIP-IQA-Q. We compare against three CLIP-IQA prompts: Complexity, Noisiness, and Quality denoted as -C, -N, -Q respectively.

Evaluators are globally sourced and must pass a set of validation questions designed to assess annotation quality. Only those who successfully complete this qualification step are allowed to rate the models.

ELO scores are computed using the same approach as SDXL-Turbo (Sauer et al., 2024), with K = 32 K=32. We find that this value of K enables more noticeable ranking adjustments, especially when models have similar performance levels.

All models in our study are evaluated using 1,000 prompts sampled from the MJHQ-30k dataset. We release this subset, which we call the `Q-Sched` split, to enable consistent benchmarking of future quantization methods. Each prompt is evaluated by four unique annotators. Therefore, each 1-vs-1 comparison results in 4,000 total human annotations.

## C  COMPUTE RESOURCES & STATISTICAL SIGNIFICANCE

We conduct all our experiments on a high-end AI server with eight Nvidia A6000s. Each model can be run independently on one A6000 and `Q-Sched` takes approximately twenty minutes to run the full grid search.

Our main experiments are averaged over two-three runs but we do not report error bars at this time.

## D  ABLATION STUDIES

### D.1  COMPARING LOSS FUNCTIONS FOR `Q-SCHED`

To evaluate the overall image quality for text-image generative modeling, CLIPScore (Hessel et al., 2021) is specifically designed to capture text-image compatibility and does not consider overall image quality. In Figure 7, we illustrate that `Q-Sched` optimized with CLIPScore produces an updated noise schedule that is over saturated and lacks image depth. In contrast, Brisque (Mittal et al., 2012) is often used as a standard reference-free image quality metric, but when used in `Q-Sched` it creates images with smoother and less detailed features. We consider three variants of CLIP-IQA (Wang et al., 2023b) and find that CLIP-IQA using the predefined quality prompt (we denote this version by CLIP-IQA-Q) achieves a noise schedule with high-fidelity images. However, CLIP-IQA-Q has a significant weakness: it cannot properly score images with hallucinations because it does not have an understanding of the underlying image prompt or concept. Therefore, we combine the benefits of CLIPScore and CLIP-IQA-Q into the JAQ loss and find that the resulting schedule fares extremely well with respect to raw image quality as well as to concept adherence.

Table 4: Adding stochasticity and its effect on W4A8 quantization for PCM using a Stable Diffusion v1-5 backbone. We report FID on COCO-30k. The stochasticity term, $\eta$, controls the amount of added Gaussian noise. $\eta = 0$ is deterministic sampling.

| Method | $\eta =$ | | | | | |
|--------|------|------|------|------|------|------|
| | 0 | 0.1 | 0.3 | 0.5 | 0.7 | 0.9 |
| TCD | 28.70 | 24.06 | **23.44** | 22.97 | 26.74 | 22.40 |
| PTQD | 23.33 | 25.59 | 24.95 | 25.69 | 24.53 | 26.71 |
| Q-Sched | **22.24** | **19.29** | **23.44** | **19.67** | **19.46** | **17.87** |

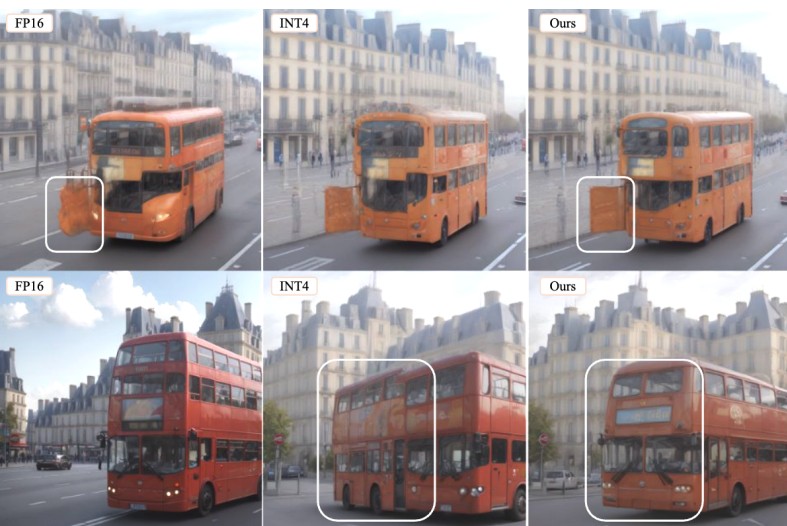

Figure 8: 4-Step (top row) and 8-Step (bottom row) LCMs. Prompt: "a car and a bus on a french highway". Q-Sched is capable of avoiding artifacts present in the FP16 or INT4 generative images. Q-Sched is close to the original schedule since it generates similar images yet our optimized schedule allows for Q-Sched to avoid some artifacts generated from the original schedule.

## D.2 ADDING STOCHASTICITY

Phased Consistency Model's implementation of the original sampler, TCD, is deterministic, meaning that there is no additive noise during sampling. The controllable noise parameter, $\eta$, allows a practitioner to adjust the additive noise during the sampling process and is defined in Equation (2). In order to compare PTQD's correction to our method, we ablate across different levels of stochasticity and report performance for six stochasticity levels in Table 4. $\eta = 0$ refers to deterministic sampling and PTQD's uncorrelated noise correction is not used since it adds stochastic noise by construction. Please see the appendix for more details on PTQD's implementation in both deterministic and stochastic sampling regimes.

We find that Q-Sched outperforms PTQD for all stochasticity regimes on the 2-step phased consistency model. With a simple grid search using our JAQ loss, we can outperform PTQD and the original TCD scheduler in different sampling regimes.

## E QUANTIZATION-INDUCED ARTIFACTS

As shown in Figure 8, Q-Sched is able to generate images that differ sufficiently from the full precision model. We ground our quantized diffusion model with image quality metrics, rather than it's error with respect to full preicision.

In our preliminary analysis using a two-step Consistency Model, we observed several characteristic ways in which quantization degrades image quality. As shown in Figure 9, quantized models tend to exhibit three prominent types of artifacts: color distortion, image degradation, and hallucinated

(a) Color Distortion | (b) Degradation | (c) Hallucinations

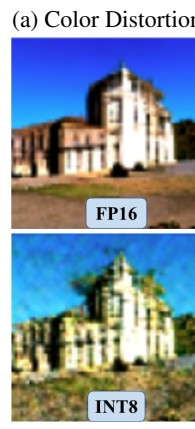 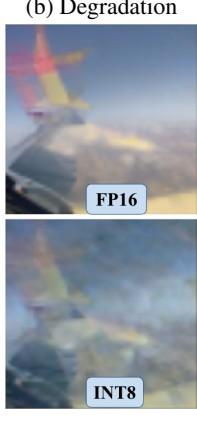 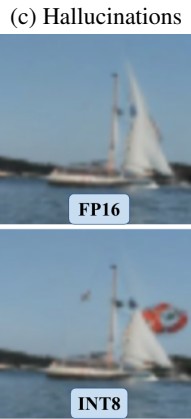

Figure 9: Three types of image artifacts that occur when quantizing image generation models. Images are unconditionally generated from a Two-Step Consistency Model (Song et al., 2023).

structures. These issues are especially pronounced in low-bit settings and appear consistently across a variety of models and prompts.

## F    LATENCY AND MODEL SIZE ANALYSIS

In Table 5, we show the full model size breakdown for the diffusion model backbone, text encoders, and the VAE decoder. During inference, either one or both text encoders are used, and we do not need the VAE encoder, since this is for training exclusively.

Table 5: FP16 Diffusion Model Size Breakdown (in GB)

|  | LCM | PCM | SDXL-Turbo | FLUX.1[schnell] |
|---|---|---|---|---|
| UNet/DiT | 1.72 | 4.84 | 1.03 | 4.76 |
| Text Encoder(s) | 0.25 | 0.29 | 0.33 | 1.95 |
| VAE Decoder | 0.07 | 0.13 | 0.02 | 0.02 |
| Total | 2.04 GB | 5.26 GB | 1.37 GB | 6.73 GB |

For our ELO *vs.* Model Size Pareto front in Figure 3, we consider the DiT memory and compute model size by taking the parameter count and multiplying it by the number of bytes required per parameter. For W4A4 + LoRA 64, the setup used for SVDQuant (Li et al., 2025), we compute the number of LoRA parameters using the back-of-the-envelope calculation provided in SVDQuant and add it to this calculation. We provide raw data for clarity in  Table 6.

Table 6: DiT Memory (in GB) for various bitwidths.

| Precision | SDXL-Turbo | FLUX.1[schnell] |
|---|---|---|
| FP16 | 1.03 | 4.76 |
| W8A8 | 0.51 | 2.38 |
| W4A4 + LoRA 64 | 0.28 | 1.24 |
| W4A8 | 0.26 | 1.19 |

In Table 7, we provide latency on an RTX A6000 on all models following the same benchmarking procedure as in Table 1.

Prior work has already benchmarked latency for SVDQuant on FLUX.1[schnell] and MixDQ on SDXL-Turbo. Here, we directly summarize those experiments. In Tables 8 and 9, INT4 and INT8 are the models that we apply `Q-Sched` with no additional overhead.

Table 7: Latency on Nvidia RTX A6000 in milliseconds. All models are evaluated in their 4-step setting.

|  | FLUX.1 | SDXL | SDv1.5 | SDXL-Turbo |
|---|---|---|---|---|
| bfloat16 | 3.881 | 0.640 | 0.193 | 0.252 |
| w4a8 | 3.372 | 0.625 | 0.172 | 0.190 |
| w4a8 `Q-Sched` | 3.256 | 0.621 | 0.172 | 0.191 |

Table 8: Reported in SVDQuant (Li et al., 2025) and summarized here for easy reference. All measurements are for FLUX.1[schnell] on an RTX 4090.

| Method | Latency (ms) |
|---|---|
| BF16 | 657 |
| INT8 | 282 |
| INT4 | 212 |
| SVDQuant | 250 |
| SVDQuant + Nunchaku | 218 |

Table 9: Latency for SDXL-Turbo on RTX 4080. These results are reported in MixDQ (Zhao et al., 2024a) and summarized here for easy reference.

| Method | Latency (ms) |
|---|---|
| BF16 | 24 |
| INT8 | 16 |

## G ADDITIONAL ANALYSIS ON COCO-30K

This result reinforces the core finding of our paper: quantization, when paired with a scheduler designed to account for noise sensitivity (as in `Q-Sched`), can be synergistic with few-step diffusion rather than detrimental. Notably, our quantized model achieves a lower FID than the original full-precision model, suggesting that `Q-Sched` helps overcome limitations introduced by both step reduction and bit-level compression.

These findings complement the results on SDXL-Turbo and FLUX.1[schnell] discussed in the main paper, and further establish `Q-Sched` as a general-purpose solution for high-fidelity, compressed diffusion generation.

## H APPLYING PTQD TO THE TCD SCHEDULER

Using PTQD's linear parameterization for the quantization error, we substitute $\mathcal{E}_\theta^Q(x_t, t) = (1 + \gamma) \cdot \mathcal{E}_\theta + \delta$ into Equation (2):

$$\mathbf{x_s} = \frac{\alpha_s}{\alpha_{s'}} \left( \alpha_{s'} \frac{\mathbf{x_t} - \sigma_t \mathcal{E}_\theta(x_t, t)}{\alpha_t} + \sigma_{s'} \mathcal{E}_\theta(x_t, t) \right) + \frac{\alpha_s}{\alpha_{s'}(1 + \gamma)} (\sigma_{s'} - \frac{\alpha_{s'} \sigma_t}{\alpha_t}) \delta + \sqrt{1 - \frac{\alpha_s^2}{\alpha_{s'}^2}} \mathbf{z}. \quad (6)$$

PTQD assumes the uncorrelated noise is sampled from a normal distribution $\delta \sim N(\mu_\delta, \sigma_\delta)$. This method applies bias correction to handle the mean deviation, $\mu_\delta$, and analytically compute standard deviation, $\sigma_\delta$. We adapt PTQD's approach to the TCD schedule and use the new standard deviation, $\sigma_\delta$ for sampling $\delta$:

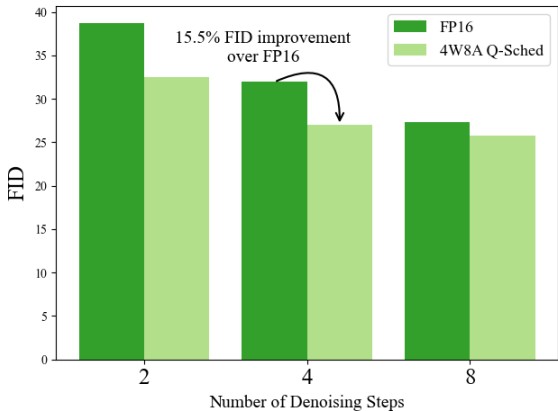

Figure 10: FID on COCO-30k. A W4A8 compressed model with our `Q-Sched` scheduler outperforms its FP16 counterpart with a $4\times$ reduction in model size.

$$\sigma_\delta^2 = 1 - \frac{\alpha_s^2}{\alpha_{s'}^2}\left(1 - \left(\frac{\delta(\sigma_{s'} - \frac{\alpha_{s'}\sigma_t}{\alpha_t})}{(1+\gamma)}\right)^2\right). \qquad (7)$$

For the edge case, where $\sigma_\delta < 0$, the deviation is set to zero ($\sigma_\delta = 0$). The proof for extending PTQD to the TCD scheduler is in the appendix.

PTQD attempts to model the distribution shift from a full precision to quantized model using two assumptions:

1. The quantized model's distribution shift can be modeled through a linear correction term.

2. The uncorrelated quantization noise is normally distributed.

While these assumptions are similar to prior work on diffusion models, they are likely to break down on the few-step diffusions where the denoising process is distilled from many steps and is not expected to be linear nor follow a Gaussian distribution.

**Quantization Noise Correction using PTQD**    Based on the PTQD quantization noise assumption, the quantization error is linearly parametrized as $\Delta\mathcal{E}_\theta = \gamma \cdot \mathcal{E}_\theta + \delta$ where $\gamma, \delta$ are learnable parameters corresponding to the correlated noise w.r.t. full precision and the uncorrelated noise respectively. PTQD models the uncorrelated noise as Gaussian (*i.e.*, $\delta \sim \mathcal{N}(\mu_q, \sigma_q)$).

**Variance Schedule Calibration for Trajectory Consistency Distillation (TCD)**    TCD's Strategic Stochastic Sampling (SSS) using a quantized network $\mathcal{E}_\theta^Q(x_t, t)$ is given by:

$$\mathbf{x_s} = \frac{\alpha_s}{\alpha_{s'}}\left(\alpha_{s'}\frac{\mathbf{x_t} - \sigma_t\mathcal{E}_\theta^Q(x_t,t)}{\alpha_t} + \sigma_{s'}\mathcal{E}_\theta^Q(x_t,t)\right) + \sqrt{1 - \frac{\alpha_s^2}{\alpha_{s'}^2}}\mathbf{z} \qquad (8)$$

Using PTQD's linear parametrization for the quantization error, we substitute $\mathcal{E}_\theta^Q(x_t, t) = (1 + \gamma) \cdot \mathcal{E}_\theta + \delta$:

$$\mathbf{x_s} = \frac{\alpha_s}{\alpha_{s'}}\left(\alpha_{s'}\frac{\mathbf{x_t} - \sigma_t((1+\gamma)\cdot\mathcal{E}_\theta(x_t,t) + \delta)}{\alpha_t} + \sigma_{s'}((1+\gamma)\cdot\mathcal{E}_\theta(x_t,t) + \delta)\right) + \sqrt{1 - \frac{\alpha_s^2}{\alpha_{s'}^2}}\mathbf{z}$$

(9)

$$= \frac{\alpha_s}{\alpha_{s'}}\left(\alpha_{s'}\frac{\mathbf{x_t} - \sigma_t(1+\gamma)\mathcal{E}_\theta(x_t,t)}{\alpha_t} + \sigma_{s'}(1+\gamma)\mathcal{E}_\theta(x_t,t) - \frac{\alpha_{s'}\sigma_t\delta}{\alpha_t} + \sigma_{s'}\delta\right) + \sqrt{1 - \frac{\alpha_s^2}{\alpha_{s'}^2}}\mathbf{z}$$

(10)

$$= \frac{\alpha_s}{\alpha_{s'}}\left(\alpha_{s'}\frac{\mathbf{x_t} - \sigma_t(1+\gamma)\mathcal{E}_\theta(x_t,t)}{\alpha_t} + \sigma_{s'}(1+\gamma)\mathcal{E}_\theta(x_t,t)\right) + \frac{\alpha_s}{\alpha_{s'}}(\sigma_{s'} - \frac{\alpha_{s'}\sigma_t}{\alpha_t})\delta + \sqrt{1 - \frac{\alpha_s^2}{\alpha_{s'}^2}}\mathbf{z}$$

(11)

The correlated noise can be corrected by applying:

$$\frac{\mathcal{E}_\theta^Q(x_t,t)}{1+\gamma} = \frac{(1+\gamma)\mathcal{E}_\theta(x_t,t) + \delta}{1+\gamma}$$

(12)

$$= \mathcal{E}_\theta(x_t,t) + \frac{\delta}{1+\gamma}$$

(13)

The resultant SSS sampling step becomes:

$$\mathbf{x_s} = \frac{\alpha_s}{\alpha_{s'}}\left(\alpha_{s'}\frac{\mathbf{x_t} - \sigma_t\mathcal{E}_\theta(x_t,t)}{\alpha_t} + \sigma_{s'}\mathcal{E}_\theta(x_t,t)\right) + \frac{\alpha_s}{\alpha_{s'}(1+\gamma)}(\sigma_{s'} - \frac{\alpha_{s'}\sigma_t}{\alpha_t})\delta + \sqrt{1 - \frac{\alpha_s^2}{\alpha_{s'}^2}}\mathbf{z}$$

(14)

The variance schedule becomes:

$$\sigma_\delta^2 = 1 - \frac{\alpha_s^2}{\alpha_{s'}^2} - \left(\frac{\alpha_s}{\alpha_{s'}(1+\gamma)}(\sigma_{s'} - \frac{\alpha_{s'}\sigma_t}{\alpha_t})\right)^2\delta^2$$

(15)

$$= 1 - \frac{\alpha_s^2}{\alpha_{s'}^2}\left(1 - \frac{(\sigma_{s'} - \frac{\alpha_{s'}\sigma_t}{\alpha_t})^2}{(1+\gamma)^2}\delta^2\right)$$

(16)

$$= 1 - \frac{\alpha_s^2}{\alpha_{s'}^2}\left(1 - \left(\frac{\delta(\sigma_{s'} - \frac{\alpha_{s'}\sigma_t}{\alpha_t})}{(1+\gamma)}\right)^2\right)$$

(17)

Since $\mathbf{z} \sim N(\mu_\delta, \sigma_\delta)$, we must handle the edge case when $\sigma_\delta < 0$. If the variance is negative, we simply set $\sigma_\delta = 0$.

Upon comparing Q-Sched to PTQD you may ask "*Why is Q-Sched able to learn a better noise schedule when it is also a linear correction?*" Q-Sched learns scalar coefficients on $x_t$ and $\mathcal{E}_\theta$ that are optimized with respect to the reference-free JAQ loss. This allows us to learn a new schedule with linear corrections to improve our overall noise schedule, rather than matching the existing full precision schedule. This is an important distinction from PTQD, which tries to learn a linear correction with respect to full precision, which may not be possible since quantization produces a nonlinear distortion on the diffusion model. In short, PTQD attempts to match the full precision sampling trajectory, whereas Q-Sched aims to learn a new sampling trajectory given a compressed $\mathcal{E}_\theta$.

# I    PROOF OF THEOREM 1: STRICT EXISTENCE GUARANTEES FOR QUANTIZATION-AWARE SCHEDULING

**Theorem 1** (Strict Existence Guarantees). *There exists Q-Sched coefficients $(\mathbf{c}^\epsilon, \mathbf{c}^\mathbf{x}) \neq 0$ such that $E[||\Delta\tilde{x}_0||] < E[||\Delta x_0||]$.*

*Proof.* Let us consider the few-step sampling trajectories for the pre-trained and quantized models, parametrized by $\mathcal{E}_\theta(t)$ and $\mathcal{E}_\theta^Q(t)$ respectively. These two few-step diffusion models sample at the same time-steps, $0 = t_0 < t_1, t_2 \cdots t_N = T$, where $N$ represents the number of steps in the few-step model. For ease of notation, we will use the time-step 0 to refer to $t_0$ and 1 to refer to $t_1$, *etc.* A denoising step going from time $t + 1 \rightarrow t$, produces a partially denoised image, $x_t$, and its quantized counterpart, $x_t^Q$. Following directly from Equation 9, the denoising error, $\Delta x_t = x_t - x_t^Q$, can be explicitly computed as:

$$\Delta x_t = \frac{\alpha_t}{\alpha_{t'}}\left(\alpha_{t'}\frac{\Delta x_{t+1} - \sigma_{t+1}(\mathcal{E}_\theta(t+1) - \mathcal{E}_\theta^Q(t))}{\alpha_{t+1}} + \sigma_{t'}(\mathcal{E}_\theta(t+1) - \mathcal{E}_\theta^Q(t+1))\right) \quad (18)$$

$$= \frac{\alpha_t}{\alpha_{t+1}}\Delta_{t+1} + \frac{\alpha_t}{\alpha_{t'}}(\sigma_{t'} - \frac{\sigma_{t+1}}{\alpha_{t+1}})(\mathcal{E}_\theta(t+1) - \mathcal{E}_\theta^Q(t+1)) \quad (19)$$

$$= k_t\Delta x_{t+1} + m_t\Delta\mathcal{E}_\theta(t+1)) \quad (20)$$

where we define the sampler coefficients as $k_t = \frac{\alpha_t}{\alpha_{t+1}}$, $m_t = \frac{\alpha_t}{\alpha_{t'}}(\sigma_{t'} - \frac{\sigma_{t+1}}{\alpha_{t+1}})$ and denote the change in the network as $\Delta\mathcal{E}_\theta(t) = \mathcal{E}_\theta(t) - \mathcal{E}_\theta^Q(t)$. Assuming the initial denoised image is the same $(x_N = x_N^Q)$, the error in the final denoised image, $\Delta x_0$, is given by:

$$\Delta x_0 = k_0\Delta x_1 + m_0\Delta\mathcal{E}_\theta(1) \quad (21)$$

$$= k_0k_1k_2...(k_N\Delta x_N + m_{N-1}\Delta\mathcal{E}_\theta(N)) + \cdots + k_0k_1m_2\Delta\mathcal{E}_\theta(3) + k_0m_1\Delta\mathcal{E}_\theta(2) + m_0\Delta\mathcal{E}_\theta(1) \quad (22)$$

$$= \sum_{s=1}^{S}\left(\Pi_{v=0}^{s-2}k_v\right)m_{s-1}\Delta\mathcal{E}_\theta(s) \quad (23)$$

The average expected error over all images in a given dataset, $x_0 \in \mathcal{D}$, is given by:

$$E[||\Delta x_0||] = \sum_{s=1}^{S}\left(\Pi_{v=0}^{S-2}k_v\right)m_{s-1}E[||\Delta\mathcal{E}_\theta(s)||] \quad (24)$$

since $E[||\Delta x_0||]$ is a homogeneous function.

In Q-Sched, we apply our quantization-aware pre-conditioning on every noise coefficient: $\tilde{m}_t = c_t^\epsilon \cdot m_t$ and $\tilde{k}_t = c_t^x \cdot k_t$. Let us denote the expected error induced by Q-Sched with respect to the pre-trained model's $x_0$ as $E[||\Delta\tilde{x}_0||]$.

We empirically show in Tables 1 and 2 that $E[||\Delta x_0||] \neq 0$ since the images produced by the naive quantization method produce a different FID from the original pre-trained model's image distribution. Since Equation 24 is a linear function of $k_t, m_t, \forall t \in 1\ldots N$, and there is a global minimum at $E[||x_0 - x_0||] = 0$, it must be that $\exists\tilde{m}_t^*, \tilde{k}_t^*\forall t$ such that $E[||\Delta\tilde{x}_0||] < E[||\Delta x_0||]$. In short, we guarantee that there exists quantization-aware coefficients that strictly improve our expected quantization error over naive quantization. $\square$

### I.1 Aside: Positive Sampler Coefficients

The TCD Scheduler has $\beta_0 = 0.0085, \beta_N = 0.012, \alpha_t = 1 - \beta_t, \sigma_t = \Pi_{i=0}^t\alpha_i$ with a scaled linear schedule:

$$\beta_t = \left(\sqrt{\beta_0} + t \cdot (\sqrt{\beta_N} - \sqrt{\beta_0})\right)^2 \quad (25)$$

Therefore: $1 > \alpha_0 > \alpha_1 > \cdots > \alpha_N > 0$ and $1 > \sigma_0 > \sigma_1 > \ldots\sigma_N > 0$. We note the $t' = (1 - \gamma)t$ where $\gamma \in [0, 1]$, so $t' \leq t$. This implies that $\sigma_{t'} > \sigma_{t+1}$ so:

$$k_t > 0 \qquad , \qquad m_t = \frac{\alpha_t}{\alpha_{t'}}(\sigma_{t'} - \frac{\sigma_{t+1}}{\alpha_{t+1}}) > 0 \qquad (26)$$

This illustrates that $k_t, m_t \in \mathbb{R}^+$.

## J  SCHEDULER COEFFICIENTS AFFECT FINAL IMAGE QUALITY

Two coefficients are expressive enough for modest compression scenarios where quantization causes a mild drift in the sampling trajectory. Adjusting the relative contribution of the predicted noise versus the intermediate state is typically enough to realign this trajectory. We observe catastrophic error when quantizing further to 3-bit integer weights and in contrast, observe only modest gains in the 8-bit scenario across both SDXL-Turbo and FLUX.1 [schnell]. We observe that Q-Sched works best in the 4-bit setting, where compression yields noticeable degradation but is not as effective in the 3-bit setting, where the generated images are very poor quality.

In Figure 11, we illustrate how three different coefficient combinations can yield images with unique artifacts and details. We take five images generated from FLUX.1[schnell] with different sampler coefficients parameterized by $(c_{min}, c_{max})$.

## K  ABLATION ON `TC()` *vs.* `IQ()`

In Table 10, we show that optimizing `Q-Sched` with respect to `TC()` or `IQ()` alone improves some metrics but not others. For example, optimizing purely for text compatibility with AQ-MAP yields strong CLIP Score and Pick Score, but noticeably lower CLIP-IQA and HPV2. In contrast, optimizing with JAQ produces consistent gains across both image quality and text compatibility. We see improvements in CLIP Score and CLIP-IQA while still maintaining strong performance on the text compatibility metrics.

| Metric | AQ-MAP | HPV2 | JAQ |
|---|---|---|---|
| CLIP Score | 25.266 | 25.043 | **25.269** |
| CLIP-IQA | 0.698 | 0.705 | **0.707** |
| HPV2 | 0.271 | **0.278** | 0.272 |
| AQ-MAP | **0.701** | 0.696 | 0.700 |
| JAQ Loss | 1.673 | 1.669 | **1.673** |
| Pick Score | **18.50** | 18.38 | 18.46 |

Table 10: Comparison of scheduler objectives across metrics.

## L  SCALABILITY TO OTHER TYPES OF DIFFUSIONS

In Table 1, we display scalability across 2,4, and 8-step consistency models and have every indication that our method will scale to longer trajectories. Yet, we emphasize that the scope of our work is to consider few-step diffusion models, which are typically less than 8 steps.

Following the same setting as in Table 1, we also provide a single run on PCM with 16 steps, illustrating exciting performance and the potential for Q-Sched to improve on longer step regimes:

| 16-step | FID |
|---|---|
| FP16 | 19.829 |
| INT4 | 19.567 |
| INT4 Q-Sched | 15.969 |

*Why not other types of few-step models?* One popular few-step diffusion model, SANA-sprint (Chen et al., 2025), is a much smaller model (0.6B parameters) than SDXL-Turbo or FLUX.1, so quantization

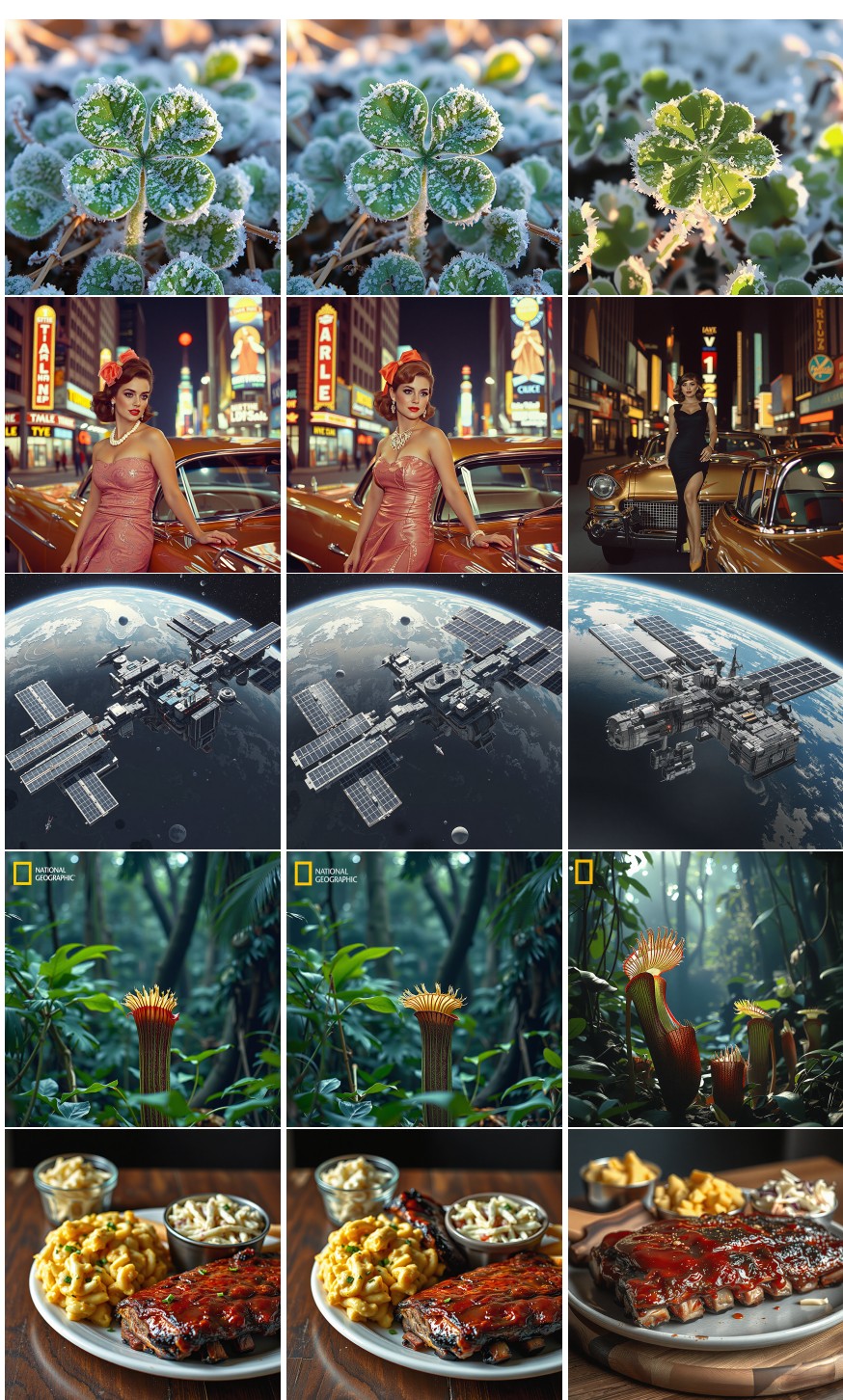

Figure 11: Qualitative comparison of generated samples across three coefficient settings. Each row corresponds to a consistent prompt, random seed and model. From left to right, coefficients $(c_{min}, c_{max})$ are $(0.8, 0.933)$, $(0.843, 0.909)$, and $(1.10, 0.82)$.

offers limited benefit and tends to cause more degradation at this scale. Q-Sched is most impactful for distilled, few-step models where size meaningfully affects inference speed, making SANA-sprint a less natural target. While Q-Sched could be applied, we leave this direction for future work.

## M GRID SEARCH DETAILS

While optimizing each $t$ independently is a reasonable search strategy, it is a significantly larger search space than our two search parameters per step ($\mathbf{c^x}$, $\mathbf{c^\epsilon}$). We apply linear spacing to avoid an exponentially large search space, which gives us excellent results.

Grid search complexity is ordinarily $O(n^\omega \cdot |\mathcal{C}|)$, where:

- $n$ is the number of search points per coefficient,
- $\omega$ is the number of diffusion steps, and
- $|\mathcal{C}|$ is the dataset size.

As the number of steps grows, ordinary grid search scales exponentially with the number of coefficients. As shown in Algorithm 1, we remove the dependency on $\omega$ by learning $(c_{\min}, c_{\max})$ and applying linear spacing for coefficients between the starting and ending timesteps. To summarize, Q-Sched scales linearly with the calibration set size and does *not* depend on the number of denoising steps.