# OpenReview forum: "Q-Sched: Pushing the Boundaries of Few-Step Diffusion Models with Quantization-Aware Scheduling"
_ICLR.cc/2026/Conference — Submitted to ICLR 2026_

### Official Review · Reviewer_aN57 · 2025-10-22

**Soundness:** 3
**Presentation:** 3
**Contribution:** 2
**Rating:** 6
**Confidence:** 3

**Summary:**

Few-step diffusion models with quantization brings both quantization error and accuracy loss due to reduced steps, causing more severe deviation in diffusion trajectories. To tackle this problem, the authors propose Q-Sched, a scheduler that learns lightweight coefficients per timestep to correct the drift between the quantized model's sampling trajectory and full precision trajectory.

**Strengths:**

1. This approach is lightweight, which aligns with the purpose of quantization and few-step diffusion.
2. The motivation is strong -- intuitively, generation quality can be much worse if combining few-step and quantization in diffusion.

**Weaknesses:**

1. Evaluation may need clarifications.
2. The methods are relatively simple.
3. Relying on a calibration set.

**Questions:**

1. Why can the FID drop below FP16 baseline by this much as Table 2(b) shows? Are the FID numbers evaluated over multiple generations? What's the error bar like?
2. If apply this paper's approach on FP16 baseline, can the FID of FP16 also be improved?
3. Are the two coefficients expressive enough? What are the limitations and under what condition do you need more coefficients or seek other approaches?
4. Do you think the calibration set can cause potential issues like overfitting?

---

> ### Author Response · Authors · 2025-11-22
> **Expressivity of Coefficients, Clarification on Table 2(b), Overfitting with Calibration Set**
>
> We thank the reviewer for their helpful feedback and recognition of our paper’s strong motivation.
>
> ***
> > [Q1] Are the two coefficients expressive enough? What are the limitations and under what condition do you need more coefficients or seek other approaches?
>
> We have updated our manuscript to include images from different coefficient permutations in Appendix J, Figure 11. These examples illustrate that different choices of $(c_{min}, c_{max})$ produce distinct visual characteristics in the generated images. For instance, the orientation and even the type of solar panel can vary noticeably across schedules.
>
> These two coefficients are expressive enough for modest compression scenarios where quantization causes a mild drift in the sampling trajectory. Adjusting the relative contribution of the predicted noise versus the intermediate state is typically enough to realign this trajectory. We observe catastrophic error when quantizing further to 3-bit integer weights and in contrast, observe only modest gains in the 8-bit scenario across both SDXL-Turbo and FLUX.1 [schnell]. We observe that Q-Sched works best in the 4-bit setting, where compression yields noticeable degradation but is not as effective in the 3-bit setting, where the generated images are very poor quality. We add this description in Appendix J.
>
> ***
> > [Q2a] Why can the FID drop below the FP16 baseline by this much as Table 2(b) shows?
>
> Because Q-Sched is optimized with respect to final output images and not dependent on intermediary features, it is likely to learn an alternative sampling trajectory from FP16. The drop in Table 2(b) indicates that the quantized model’s images more closely match the target distribution than the original model. With distilled few-step models, it is possible that Q-Sched perturbs the sampling trajectory in the correct direction to learn a better overall image distribution. However, FID is an incomplete metric and does not capture fine-grained image artifacts like other image quality metrics. In Table 3, we illustrate how different image quality metrics score image distributions differently, and Q-Sched on INT8 SDXL-Turbo (4-Steps) shows an average of 0.6% improvement across image quality metrics even though it has 3.3% degradation in FID (in Table 2b). Overall, we show that Q-Sched is competitive with respect to both state-of-the-art quantization methods and full precision. We add some intuition about this in lines 428-430.
>
> ***
> > [Q2b] Are FID numbers evaluated over multiple generations? What are the error bars like?
>
> In Table 2b, we average over three runs. Below is our standard deviation and raw numbers across seeds 40,41, and 42:
>
> | s42     | s41     | s40     | average | std_dev       |
> |---------|---------|---------|---------|----------------|
> | 25.534  | 25.398  | 25.502  | 25.478  | 0.07110555534 |
> | 25.953  | 25.431  | 25.862  | 25.749  | 0.2788446401  |
> | 25.553  | 25.264  | 25.250  | 25.356  | 0.1710389819  |
> | 21.583  | 21.291  | 21.361  | 21.412  | 0.1524510851  |
>
> We report the standard deviation in the text on lines 425-426.
>
> ***
> > [Q3] If the paper’s approach is on the FP16 baseline, can the FID of FP16 also be improved?
>
> It is possible that the FP16 baseline may be improved with our method, particularly since these are distilled models. However, this is not the use case for our method and we leave this investigation to future work.
>
> ***
> > [Q4] Do you think the calibration set can cause potential issues like overfitting?
>
> This is a concern in all calibration-based PTQ methods including popular methods such as GPTQ and SmoothQuant. Our work attempts to mitigate overfitting by (1) using a highly descriptive calibration set and (2) only optimizing a few coefficients. We use sDCI (long form) prompts for calibration and MJHQ or COCO-30k for evaluation. By using long form calibration prompts, we attempt to capture how the quantized diffusion can handle highly complex scenes, which likely will generalize well to simpler prompts. We find the sDCI prompts sufficient for evaluating overall generative image quality, and demonstrate excellent performance on MJHQ and COCO-30k, two completely different downstream datasets. We add a short discussion of our calibration set  on line 372-375.

---

> > ### Author Response · Authors · 2025-11-25
> > **Follow-Up During Discussion Period**
> >
> > Thank you again for your thoughtful review. If you have any further questions or would like clarification on any part of our response, we’d be grateful for additional feedback during the discussion period.

---

### Official Review · Reviewer_mdUQ · 2025-10-28

**Soundness:** 2
**Presentation:** 2
**Contribution:** 3
**Rating:** 4
**Confidence:** 4

**Summary:**

This paper proposes Q-Sched, a quantization-aware scheduler designed for few-step diffusion models.
Instead of modifying model weights or activations, Q-Sched adapts the sampling scheduler itself to mitigate trajectory drift caused by quantization.

**Strengths:**

1. Novel Perspective. The idea of addressing quantization artifacts at the scheduler level rather than the model level is interesting and practically elegant.

2. Experiments span multiple diffusion families (LCM, PCM, SDXL-Turbo, FLUX.1), and include both objective metrics (FID, CLIPScore) and large-scale human evaluations.

**Weaknesses:**

1. The proposed scheduler-level correction resembles existing bias/variance scaling approaches such as PTQD. The distinction between Q-Sched and these prior works is not clearly articulated.

2. Several figures are rasterized screenshots instead of scalable vector graphics, resulting in unclear visuals and text artifacts in the PDF.

3. The method’s scalability to models with longer trajectories (e.g., > 8 steps) or higher-resolution datasets is not analyzed. It remains uncertain how the grid-search complexity grows with timestep count or calibration-set size.

**Questions:**

see weaknesses

---

> ### Author Response · Authors · 2025-11-22
> **Scalability, Grid Search Complexity, and Comparison with Prior Scaling Techniques**
>
> We thank the reviewer for their helpful feedback and recognition of our novel results.
>
> ***
> > [W1] The method’s scalability to models with longer trajectories (e.g., >8 steps) or higher-resolution datasets is not analyzed.
>
> In Table 1, we display scalability across 2,4, and 8-step consistency models and have every indication that our method will scale to longer trajectories. Yet, we emphasize that the scope of our work is to consider few-step diffusion models, which are typically less than 8 steps.
>
> Following the same setting as in Table 1, we also provide a single run on PCM with 16 steps, illustrating exciting performance and the potential for Q-Sched to improve on longer step regimes. We add the following in Appendix L, Table 1:
>
> | 16-step       | FID     |
> |---------------|---------|
> | FP16          | 19.829  |
> | INT4          | 19.567  |
> | INT4 Q-Sched  | 15.969  |
>
> ***
> > [W2] It remains uncertain how the grid-search complexity grows with timestep count or calibration-set size.
>
> Grid search complexity is ordinarily $O(n^{\omega} \cdot  |\mathcal C | )$ where:
>
> * n is the number of search points per coefficient
>
> * $\omega$ is the number of diffusion steps
>
> * $|\mathcal C|$ is the dataset size
>
>
> As the number of steps grows, ordinary grid search scales exponentially with the number of coefficients. As shown in Algorithm 1, we remove the dependency on $\omega$ and instead choose to learn $(c_min, c_max)$ and apply linear spacing for coefficients between the starting and ending timesteps. To summarize, Q-Sched scales linearly with calibration set size and does not depend on the number of denoising steps. We add this description in Appendix M.
>
> ***
> > [W3] proposed scheduler-level correction resembles existing bias/variance scaling approaches such as PTQD. The distinction between Q-Sched and these prior works is not clearly articulated.
>
> While there are superficial similarities between scheduler-level corrections and prior bias and variance scaling methods, the mechanisms and assumptions underlying Q-Sched differ in several important ways. The most notable distinction is that Q-Sched introduces learnable coefficients that are optimized end-to-end with respect to the final image quality, rather than relying on Gaussian priors or intermediate-step signals. Q-Sched does not assume any structure on the correction terms, nor does it depend on partially denoised images during optimization, which helps avoid introducing bias tied to intermediate distributions. This is particularly important in few-step (distilled) models where the intermediate distributions are no longer Gaussian.
> PTQD assumes that quantization error at each denoising step arises solely from the error in the current noise prediction network, i.e., the distortion in $\mathcal E_\theta^Q(x_t, t)$. Q-Sched instead applies two complementary terms: one acting on x_t​, which captures accumulated error from previous steps, and one acting on the current noise estimate $\mathcal E_\theta^Q(x_t, t)$. This separation provides additional flexibility by allowing the method to adjust both the propagated state and the current prediction independently.
> Finally, PTQD models quantization noise as a linear combination of correlated and uncorrelated Gaussian components with respect to the full-precision model, and these parameters are learned relative to the intermediate timestep distribution. Q-Sched, in contrast, optimizes its coefficients directly toward final-stage image quality, aligning the correction with the ultimate objective rather than the properties of intermediate denoising states.
> In short the key improvements in Q-Sched over PTQD are:
> * Adding a coefficient on $x_t$
> * Removing dependency on full precision activations
> * Relaxing the Gaussian noise assumption
> * Optimizing with respect to the output images rather than intermediary states
>
> We add a summary of this explanation in lines 209-216.
> ***
> > [W4] Several figures are rasterized screenshots and not svg
>
> Thank you for bringing this to our attention, we have fixed Figures 1, 2,and 5, we believe the other figures are sufficiently high resolution.

---

> > ### Author Response · Authors · 2025-11-25
> > **Follow-Up During Discussion Period**
> >
> > Thank you again for your thoughtful review. If you have any further questions or would like clarification on any part of our response, we’d be grateful for additional feedback during the discussion period.

---

> ### Author Response · Authors · 2025-11-28
> **Score Increase to Weak Accept (6)**
>
> Thank you for raising the score to weak accept (6), we would appreciate feedback on what we can improve on further.

---

### Official Review · Reviewer_pBRf · 2025-10-28

**Soundness:** 3
**Presentation:** 3
**Contribution:** 2
**Rating:** 6
**Confidence:** 5

**Summary:**

The paper studies the problem of the degradation of generation fidelity when diffusion models are applied using low quantization. The paper proposes Q-Sched, which identifies two parameters in the scheduling of the quantized diffusion steps, to mitigate the degradation from quantization. As only two parameters are required, Q-Sched utilizes grid search and proposes to find the parameter that best optimizes a JAQ loss, which is a combination of the text-image matching metric and image quality metric. In experiments, the paper shows that Q-Sched can improve the FID and CLIP score of the quantized model compared to no-modification or baseline scheduling, especially under the setting of low quantization.

**Strengths:**

1. The paper proposes a simple approach that only tunes two parameters of the diffusion schedule to improve the performance of the quantized model.

2. The paper gives intuitive theoretical justification for the design of the two parameters.

3. The proposed method improves CLIP and FID, especially under the setting of low quantization.

**Weaknesses:**

1. More qualitative comparisons or essentially visual comparisons between images generated by different methods should be provided.

2. Conceptually, the metric for the adapted schedule should be to match the output of the non-quantized model. However, the paper proposes to use the JAQ loss as the target for the grid search. Even though JAQ could be a comprehensive metric, it is not clear why it could serve as the ideal judgment for tuning the scheduling for the quantized model.

3. More details of the grid search results could be added. Since there are only two parameters, we could clearly see what the trend is and how sensitive the JAQ loss is with respect to the parameters. This could also justify why the proposed method requires few prompts for tuning to achieve good performance.

minor:
1.  line 265: a those

**Questions:**

1. What if we apply Q-Sched on the non-quantized model? Could it be the case that we see better performance because JAQ is better aligned with the metrics?

2. How to explain in Table 2 (b), Q-Sched has a quite lower FID compared to the FP16 model?

---

> ### Author Response · Authors · 2025-11-22
> **Visual Comparison, JAQ Loss Sensitivity, and Clarifications on Table 2b**
>
> We thank the reviewer for their helpful feedback and recognition of our paper’s theoretical justification.
>
> ***
> > [W1] more qualitative or visual comparisons between images is needed
>
> In Appendix A, Figure 6, we provide a side-by-side comparison of SVDQuant and Q-Sched on 10 images from the MJHQ dataset. We illustrate that Q-Sched finds a unique sampling trajectory resulting in subtle detail changes from SVDQuant which we attribute to our added coefficients in the few-step scheduler.
>
> ***
> > [W2] Conceptually, the metric for the adapted schedule should be to match the output of the non-quantized model. Even though JAQ could be a comprehensive metric, it is not clear why it could serve as the ideal judgment for tuning the scheduling for the quantized model.
>
> With distilled few step models, the partially denoised images and intermediate activations from the full precision model do not provide a stable optimization target. Quantization often introduces shifts in structure, texture, or semantics that cannot be reconciled by simply matching the full precision trajectory. For example, as shown in Figure 1, the W4A4 SVDQuant model generates a popsicle with a prominent hand holding it, while the full precision model produces far less of the hand. Minimizing the difference between these outputs forces the scheduler to track both quantization induced artifacts and the natural drift between the two models, which is not a well defined or easily expressible objective.
> In contrast, optimizing directly for downstream image quality rather than attempting to replicate the full precision outputs provides a more reliable target. JAQ is not intended to approximate the full precision denoising path but to guide the compressed model toward producing high quality images under its own generative dynamics. We explain this on lines 134-147 and 202-208.
> ***
> >  [W3] More details of the grid search results could be added. Since there are only two parameters, we could clearly see what the trend is and how sensitive the JAQ loss is with respect to the parameters. This could also justify why the proposed method requires few prompts for tuning to achieve good performance.
>
> In light of this comment, we have added visual comparisons for three representative Q-Sched coefficient settings in Appendix J, Figure 11. These examples illustrate that different choices of $(c_{min}, c_{max})$ produce distinct visual characteristics in the generated images. For instance, the orientation and even the type of solar panel can vary noticeably across schedules. This variability highlights the sensitivity of the coefficients and demonstrates why a small calibration set is sufficient: the search space is low dimensional, and the qualitative differences emerge clearly even with a few prompts.
>
> ***
> > [Q1] What if we apply Q-Sched on the non-quantized model? Could it be the case that we see better performance because JAQ is better aligned with the metrics?
>
> While it is theoretically possible that applying Q-Sched to a non quantized model could yield some improvement, this is not the setting the method is designed for. A full precision model is best optimized directly through its original training objectives such as distillation and fine-tuning rather than through a post-training scheduler adjustment.
> Q-Sched specifically targets post training degradation introduced by quantization and is intended to correct the resulting drift in the sampling trajectory. Its purpose is not to act as a general post training refinement mechanism for full precision models but to restore consistency and quality in compressed ones. We add a section about this on lines 319-323.
> ***
> > [Q2] How to explain in Table 2 (b), Q-Sched has a quite lower FID compared to the FP16 model?
>
> Because Q-Sched is optimized with respect to final output images and not dependent on intermediary features, it is likely to learn an alternative sampling trajectory from FP16. The drop in Table 2(b) indicates that the quantized model’s images more closely match the target distribution than the original model. With distilled few-step models, it is possible that Q-Sched perturbs the sampling trajectory in the correct direction to learn a better overall image distribution. However, FID is an incomplete metric and does not capture fine-grained image artifacts like other image quality metrics. In Table 3, we illustrate how different image quality metrics score image distributions differently, and Q-Sched on INT8 SDXL-Turbo (4-Steps) shows an average of 0.6% improvement across image quality metrics even though it has 3.3% degradation in FID (in Table 2b). Overall, we show that Q-Sched is competitive with respect to both state-of-the-art quantization methods and full precision. We address the dramatic FID reduction in more detail on lines 421-427.

---

> > ### Author Response · Authors · 2025-11-25
> > **Follow-Up During Discussion Period**
> >
> > Thank you again for your thoughtful review. If you have any further questions or would like clarification on any part of our response, we’d be grateful for additional feedback during the discussion period.

---

### Official Review · Reviewer_53Sq · 2025-10-31

**Soundness:** 2
**Presentation:** 3
**Contribution:** 4
**Rating:** 4
**Confidence:** 3

**Summary:**

This paper addresses the challenges to adopting quantized text-to-image diffusion models for few-step inference by proposing a novel scheduler-level PTQ approach that adapts the sampler rather than the model weights.

The key idea behind Q-Sched is to introduce two lightweight, learnable preconditioning coefficients ($c^x, c^{\epsilon}$) per timestep, which are applied to the scheduler's inputs.

To optimize these coefficients, the authors introduce the Joint Alignment-Quality (JAQ) loss, a reference-free objective that combines text-image compatibility with a pure image quality score, and apply grid search for these coefficients.

Experimental results show that Q-Sched can match or even surpass the quality of FP16 baselines (e.g., 15.5% FID improvement over 4-step LCM) and outperform other PTQ methods like MixDQ and SVDQuant in human preference.

**Strengths:**

1. The motivation of this paper that combining few-step inference and quantized model is well established and matches real-world application.
1. This proposed Q-Sched acheives promising experimental perfomance on FID improvement and CLIPScore.
1. The core idea of reframing the quantization problem as a trajectory drift issue and solving it by adapting the scheduler instead of the weights  is novel and elegant.
1. The JAQ loss is also a well-motivated and practical contribution for this reference-free optimization task.
1. The writing is clear, concise, and easy to follow. The core problem of "scheduler mismatch"  is intuitively explained and provides a strong motivation for the proposed solution.

**Weaknesses:**

1. Lack of latency comparison: one common motivation of model quantization and few step inference is to reduce model latency, however, this paper does not show comparision between Q-Sched and other models on model latency.
1. There is an ablation study that determine how to choose $k$ for the JAQ Loss, but it does not include ablation study that use pure $\text{TC}(\cdot)$ or pure $\text{IQ}(\cdot)$, or-not reference-free loss.
1. The Algorithm 1 should be a grid search algorithm that uses $c_{min}$ and $c_{max}$, but it is instead an unexplained search optimizer `opt` with `opt.step`.
1. The Q-Sched is well motivated and justificated by math, but it misses the intuition why Equation (4) is designed as such a form.

**Questions:**

1. In Algorithm 1, it seems that the $(\mathbf{c^x}, \mathbf{c^\epsilon}):=(c_t^x, c_t^\epsilon)_{t=0}^T$ are optimized simutenously. How about optimizing each $t$ independently (or they have to be linspaced)?
1. Why $\sigma_{s'}$ is combined with $c_t^\epsilon$ but not $c_{s'}^\epsilon$? Any justification?
1. I notice there is a model named SANA-sprint that proposes a training-free approach that transforms a pre-trained flow-matching model for continuous-time consistency distillation (sCM), any comparision with it and Q-Sched?
1. Is the grid search of Q-Sched (Algorithm 1) faster than full-presicion calibration used by other methods?

---

> ### Author Response · Authors · 2025-11-22
> **Ablation, Latency Comparison, and Clarity on Algorithm 1**
>
> We thank the reviewer for their helpful feedback and recognition of our exciting experimental results.
> ***
> > [W1] Ablation study that use(s) pure TC() or pure IQ() or not reference-free loss
>
> In Appendix K, Table 9, we include this ablation to show that optimizing Q-Sched with respect to TC() or IQ() alone improves some metrics but not others. For example, optimizing purely for text compatibility with AQ MAP yields strong CLIP Score and Pick Score, but noticeably lower CLIP IQA and HPV2. In contrast, optimizing with JAQ produces consistent gains across both image quality and text compatibility. We see improvements in CLIP Score and CLIP IQA while still maintaining strong performance on the text compatibility metrics.
>
> |Metric|AQ-MAP|HPV2|JAQ|
> |--|--|--|--|
> |CLIP Score|25.266|25.043| **25.269**|
> |CLIP-IQA|0.698|0.705| **0.707**|
> |HPV2|0.271| **0.278**|0.272|
> |AQ-MAP|**0.701**|0.696|0.700|
> |JAQ Loss|1.673|1.669|**1.673**|
> |Pick Score|**18.50**|18.38|18.46|
> ***
> > [W2]  Lack of latency comparison
>
> In Appendix F, we add the latency measurements reported in SVDQuant for FLUX.1[schnell] on an RTX 4090:
> |Method|Latency (ms)|
> |--|--|
> |BF16 | 657|
> |INT8 | 282|
> |INT4| 212|
> |SVDQuant |250|
> |SVDQuant + Nunchaku|218|
>
> We focus on INT4 models, where Q-Sched is compared directly against SVDQuant, which has an FP16 adapter, under the same quantization setting. As shown in Figure 1, Q-Sched reaches a better latency-output quality trade-off, achieving competitive ELO scores without adding computational overhead as shown in Figure 1.
>
> MixDQ also reports INT8 latency on a RTX 4080:
> |Method|Latency (ms) |
> |--|--|
> |BF16|24|
> |INT8|16|
>
> In this setting, Q Sched optimizes the schedule of the INT8 model and achieves state-of-the-art image quality while operating at the same latency as MixDQ as reported in Table 7.
>
> ***
> > [W3] The Algorithm 1 should be a grid search algorithm but it is instead an unexplained search optimizer opt with opt.step
>
> In Algorithm 1, we use a generic optimizer to indicate that different search strategies are possible. In our experiments, we instantiate a grid search for the optimizer, and `opt.step` simply advances to the next point in the predefined search grid. If one were to use gradient descent or another continuous optimizer, `opt.step` would instead update the coefficients based on the corresponding rule.
>
> We present the procedure in a general form and explain on lines 253-258 that grid search is the specific optimizer we use. We find grid search sufficient and reliable for this problem, but the algorithm is written to remain compatible with alternative optimization strategies.
>
> ***
> > [W4] misses the intuition why Equation (4) is designed as such a form
>
> Q-Sched introduces two scalar coefficients that act on these components as already described on lines 238-240. Equation 4 is written to make explicit how Q Sched interacts with the update rule of a common few-step sampler, specifically the TCD scheduler used in PCMs. The TCD update computes two key quantities: the model’s noise prediction at timestep t, $\mathcal E_\theta^Q(x_t, t)$, and an auxiliary prediction evaluated at an intermediate timestep $s'$. Together, these terms parameterize the score function that advances the sample from t to s. Q-Sched coefficients can be fused into the existing TCD coefficients without modifying the computational graph or adding inference cost.
> ***
> > [Q2] Why is $\sigma_{s’}$ combined with $c_t^\epsilon$ but not $c_{s’}^{\epsilon}$?
>
> In the TCD formulation, the estimate at the proxy timestep s′ is not obtained from a separate model evaluation but is derived directly from the prediction at timestep t as shown in Equation 2 and highlighted on lines 226-227. As a result, the noise term associated with s′  inherits the same coefficient $c_t^\epsilon$.
>
> The coefficient $c_{t}^{\epsilon}$ is applied to the model output, $\mathcal E_\theta (x_t, t)$, and is fused to *both* $\sigma_t$ and $\sigma_{s’}$ during the prediction of $x_s$.
>
> ***
>
> > [Q1] In Algorithm 1, (coefficients) are optimized simultaneously. How about optimizing each t independently?
>
> While optimizing each t independently is a reasonable search strategy, it is a significantly larger search space than our two search parameters per step $(\mathbf{c^x}, \mathbf{c^\epsilon})$. We apply linspace to avoid an exponentially large search space, which gives us excellent results. This description is added in Appendix M.
>
> ***
> > [Q3] Any comparison with Q-Sched on SANA-sprint?
>
> SANA-sprint has only 0.6B parameters, compared to 0.9B and 12B for SDXL Turbo and FLUX.1[schnell]. At this scale, quantization tends to introduce larger degradation with limited speedup, since the model is already small enough for most use cases. In general, Q Sched is most effective for distilled few step models where model size meaningfully limits inference speed. While Q Sched could be applied to SANA sprint, we leave this exploration to future work. This description is added in Appendix L.

---

> > ### Author Response · Authors · 2025-11-25
> > **Follow-Up During Discussion Period**
> >
> > Thank you again for your thoughtful review. If you have any further questions or would like clarification on any part of our response, we’d be grateful for additional feedback during the discussion period.

---

> ### Comment · Reviewer_53Sq · 2025-11-26
>
> Thanks for your replies. I still have some concerns.
>
> > [W2] Lack of latency comparison
>
> You should me a table about the latency of some quantization. But I want to compare full-precision model vs quantized model vs. few-step model vs Q-Sched model regarding the latency and performance (I expected to have another latency column in Table 1).
>
> > [Q3] Any comparison with Q-Sched on SANA-sprint?
>
> What I want to know is that small few-step model like SANA-sprint is small enough to use, so do we still need to further apply Q-Sched for quantization? And your answer is no:
> > the model is already small enough for most use cases
>
> This lets me doubt how much Q-Sched can contribute to real applications. Thus, I want to see the comparison between Q-Sched models and small few-step models like SANA-sprint, with respect to latency and performance.

---

> > ### Author Response · Authors · 2025-12-03
> > **Response to Latency Concerns & Comparison with SANA-Sprint**
> >
> > We sincerely thank the reviewer for their recognition that our motivation is well established and matches real-world applications.
> >
> > ***
> > > [W2] Lack of latency comparison, I expected to have another latency column in Table 1
> > Per the reviewer’s request, we now include a latency comparison in Table 1. We benchmarked all models on an NVIDIA RTX A6000 and added a detailed description of our measurement procedure in the experimental setup section. A summary of the results is shown below:
> >
> > Latent Consistency Models (with SDv1.5 backbone) across different few-step regimes (2-step, 4-step, etc.)
> > | Method|2| 4| 8| 16|
> > |--|--|--|--|-|
> > | bfloat16| 0.148 | 0.193 | 0.286 | 0.466 |
> > | w4a8|0.136| 0.172 | 0.245 | 0.393|
> > | w4a8 PTQD| 0.137 | 0.172 | 0.246 |0.398|
> > | w4a8 Q-Sched|0.136|0.172|0.245|0.396|
> > These results confirm that Q-Sched achieves iso-latency and iso-throughput relative to standard INT4 quantization. The Q-Sched coefficients replace the existing scheduler parameters and incur no additional operations or kernel launches, so inference cost is identical to the underlying quantized model.
> > Minor numerical differences between INT4 and INT4+Q-Sched arise only from measurement noise in microbenchmarking, not from any added overhead. Thus, Q-Sched preserves the full efficiency of INT4 inference while improving image quality.
> >
> > ***
> >
> > > I want to know is that small few-step model like SANA-sprint is small enough to use, so do we still need to further apply Q-Sched for quantization? And your answer is no
> >
> > First, SANA-sprint was published after ICLR’s deadline and therefore should be considered concurrent work.
> >
> > Second, our answer is not “no.” SANA-Sprint is an atypically small backbone, so it is not a good target for quantization. As stated in our response, Q-Sched is most effective for distilled few-step models where “model size meaningfully limits inference speed”. In the case of SANA-Sprint, the model is ~0.6 GB total, of which 500 MB is the text encoder. The transformer backbone, which is the component targeted by Q-Sched, is therefore only about 100 MB. Because the backbone is already so small, quantization provides limited practical acceleration, making SANA-Sprint an unrepresentative setting for evaluating the benefits of Q-Sched. Q-Sched is important and useful, but its impact is most visible for larger few-step models with full DiT backbones, where model size is a real performance bottleneck (see Table 5 and below for Memory Breakdown).
> >
> > **FP16 Diffusion Model Size Breakdown (in GB)**
> > | Component | LCM | PCM | SDXL-Turbo | FLUX.1[schnell] |
> > |--|--|--|--|--|
> > | UNet/DiT| 1.72 | 4.84 | 1.03 | 4.76 |
> > | Text Encoder(s)| 0.25 | 0.29 | 0.33 | 1.95 |
> > | VAE Decoder| 0.07 | 0.13 | 0.02 | 0.02 |
> > | Total| 2.04 GB | 5.26 GB | 1.37 GB | 6.73 GB |
> >
> >
> > ***
> > > lets me doubt how much Q-Sched can contribute to real applications.
> >
> > To address this concern about Q-Sched’s impact in practical settings, we include an expanded latency evaluation in Appendix F (Table 7), reporting end-to-end latency for all models in the paper using the same protocol as Table 1. These results show that Q-Sched provides consistent reductions in inference latency compared to the original bfloat16 few-step models. For example, INT4 quantization (with Q-Sched) reduces FLUX.1 latency from 3.881 ms → 3.256 ms (≈16% reduction).  As summarized below, the latency improvements appear across architectures and step counts, indicating that Q-Sched offers clear benefits in real applications.
> >
> > | Method        | FLUX.1 | SDXL | SDv1.5 | SDXL-Turbo |
> > |---------------|--------|------|--------|------------|
> > | bfloat16      | 3.881  | 0.640 | 0.193 | 0.252 |
> > | w4a8          | 3.372  | 0.625 | 0.172 | 0.190 |
> > | w4a8 Q-Sched  | 3.256  | 0.621 | 0.172 | 0.191 |
> >
> > ***
> >
> > > I want to see the comparison between Q-Sched models and small few-step models like SANA-sprint, with respect to latency and performance.
> >
> > We now include end-to-end latency measurements and performance comparisons in Table 1, addressing the reviewer’s request. Table 7 reports latency for all remaining models. Across these results, Q-Sched consistently delivers state-of-the-art image quality, supported by human preference studies (Figs. 3–4) and a broad suite of automated metrics (Tables 1–3). Together, these experiments provide a thorough evaluation of Q-Sched across both latency and quality, showing clear advantages over existing baselines.
> > Regarding SANA-Sprint, we clarify that **it appeared after the ICLR deadline and is therefore concurrent**. More importantly, it is a very small model (~600 MB total, ~100 MB transformer), fundamentally different from few-step DiT-based models such as SDXL-Turbo and FLUX.1[schnell]. Quantizing large DiTs is technically challenging and crucial for real-world acceleration, and SANA-Sprint is not a representative setting for evaluating Q-Sched, whose strengths are most impactful on high-capacity DiT backbones where quantization meaningfully affects the speed–quality tradeoff.

---

### Author Response · Authors · 2025-11-22
**Overview of added ablations, latency baselines, algorithm clarifications, and visual/grid-search analyses.**

Across all reviews, we appreciate the recognition of our contributions and have expanded the paper to address each concern in detail. We have added missing ablations, clarified the role and implementation of Algorithm 1, provided latency comparisons against SVDQuant and MixDQ, and included new qualitative visualizations to illustrate coefficient behavior and schedule sensitivity. We also strengthened our explanation of Equation 4 and the interaction between Q-Sched and the TCD update, and clarified why JAQ is used as the optimization target for quantized few-step models. To address scalability and complexity questions, we now explicitly analyze grid-search cost, show that Q-Sched scales independently of timestep count, and add a 16-step PCM result demonstrating promising performance beyond the few-step regime. We further distinguish Q-Sched from PTQD and other bias and variance scaling approaches by detailing differences in assumptions, objectives, and correction mechanisms. Finally, we expanded discussion around expressivity, FID behavior, variance across seeds, and overfitting risks, and added additional figures and tables accordingly. Overall, we believe these additions fully resolve the reviewers’ questions and further clarify the novelty, rigor, and robustness of Q-Sched.

We thank all reviewers for their detailed feedback. Across the revised manuscript, we have addressed every comment as follows:
##  Reviewer 53Sq ( Weak Reject 4) — Ablations, Latency, Algorithm 1, Equation 4
* Added full ablations for pure TC(), pure IQ(), and non–reference-free losses (Appendix K).


* Added latency comparisons against SVDQuant, MixDQ, and BF16 baselines, with full measurement protocol (Table 1, Appendix F).


* Clarified Algorithm 1: grid search as the instantiated optimizer and how opt.step operates in this setting.


* Expanded intuition and explanation for Equation (4) and clarified its interaction with the TCD scheduler.


* Explained why $σ_s′​$ is fused to the coefficient $c_t^\epsilon$


* Justified joint optimization of coefficients and clarified search-space reduction strategy (Appendix M).
* Explained why SANA-Sprint is outside the scope of large-backbone few-step quantization (Appendix L).

## Reviewer pBRf  (Weak Accept 6) — Visual Comparisons, JAQ Justification, Grid-Search Sensitivity
* Added new qualitative side-by-side visual comparisons (Appendix A).


* Clarified why JAQ is a stable and appropriate optimization target for quantized few-step models, rather than matching FP16 intermediates.


* Added grid-search sensitivity visuals and examples showing coefficient effects (Appendix J).


* Clarified that Q-Sched targets quantization drift and is not intended to improve full-precision models.


* Explained FID behavior relative to other metrics and added discussion on metric disagreement (lines 421–427).

## Reviewer mdUQ (Weak Reject 4, **moved to Weak Accept (6) during Rebuttal period**) — Scalability, Grid-Search Complexity, Relation to PTQD
* Added a 16-step PCM result showing promising scalability beyond the <8-step regime (Appendix L).


* Analyzed grid-search complexity and showed Q-Sched scales independently of timestep count due to linear coefficient spacing (Appendix M).


* Clarified distinctions from PTQD: removal of Gaussian assumptions, independence from FP activations, additional coefficient on $x_t$​, and optimization toward final images.


* Updated rasterized figures to SVG where needed.

## Reviewer aN57 (Weak Accept 6) — Expressivity, FID Interpretation, Calibration Overfitting
* Added visuals showing expressivity of ($c_{\min}, c_{\max}$) and limitations in more extreme quantization regimes (Appendix J).


* Provided intuition for why Q-Sched’s sampling trajectory can yield lower FID than FP16 (lines 428–430).


* Reported FID variance across seeds and added standard deviations (lines 425–426).


* Discussed potential calibration-set overfitting and mitigation strategies via descriptive long-form prompts and low-dimensional coefficient search.

---

### Meta-Review · Area_Chair_5EQn · 2026-01-03

**Summary:**

This paper proposes Q-Sched, a scheduler-level post-training quantization (PTQ) method specifically designed for few-step diffusion models. Instead of modifying model weights or activations, Q-Sched learns per-timestep preconditioning coefficients that are optimized using a Joint Alignment–Quality (JAQ) loss, which is reference-free and combines image-text alignment and image quality metrics. The approach is lightweight, requiring only two parameters per step, and aims to reduce quantization-induced degradation without access to full-precision activations.

While reviewers appreciated the motivation and recognized the potential of scheduler-based PTQ for efficient diffusion, the overall reception was mixed, with several reviewers expressing concerns about novelty, evaluation completeness, and practical impact.

The main concerns included:

**Evaluation limitations**: Reviewer 53Sq requested clearer latency comparisons and a direct comparison with small few-step models like SANA-sprint, questioning whether Q-Sched was competitive in real-world low-resource settings.

**Conceptual clarity around the JAQ loss**: Reviewer pBRf questioned why the JAQ loss—rather than matching full-precision outputs—was appropriate for tuning the scheduler.

**Scalability and comparison with existing methods**: Reviewer mdUQ noted that Q-Sched resembles existing scaling approaches like PTQD, and found the distinctions underexplained in the original submission. Also raised concerns about scalability to longer trajectories and grid search complexity.

**Simplicity and reliance on calibration**: Reviewer aN57 found the method relatively simple and raised valid concerns about calibration set overfitting, coefficient expressivity, and the surprising result that Q-Sched sometimes outperforms the FP16 baseline.

Overall, while Q-Sched is a well-motivated and carefully executed piece of work, and the rebuttal was excellent, the paper remains incremental in terms of conceptual contribution. The core idea of scheduler-level correction via grid-searched coefficients is elegant but limited in scope, and the overall gains over baselines are modest. It does not introduce a fundamentally new class of methods or meaningfully advance the state of quantized diffusion beyond existing work.

**Reviewer Concerns:**

**Addressed Concerns**\
The authors provided a thorough and thoughtful rebuttal, with additional experiments, clarifications, and figures:

***Latency and real-world comparisons (53Sq)***: The authors added Table 1 and Appendix F, benchmarking Q-Sched against full-precision and other quantized models. They demonstrated iso-latency with INT4 baselines and an average 16% latency improvement over bfloat16 models.

***Comparison with small models (SANA-sprint)***: The authors clarified that SANA-sprint is concurrent work and not representative of the large DiT-based backbones where Q-Sched is most effective. They argued that quantization has limited benefit for already-small models.

***JAQ loss justification (pBRf)***: The authors elaborated that reference-based loss functions (e.g., matching FP16 trajectories) are unreliable due to quantization-induced drift. They advocated for optimizing the final image quality directly via JAQ.

***Visual expressivity and grid search (aN57)***: The authors added Appendix J with image comparisons under different coefficient settings, and described the low-dimensional, tunable nature of the coefficient space, justifying the small calibration set.

***Scalability (mdUQ)***: The authors added a 16-step result and explained how Q-Sched’s coefficient design avoids exponential growth by using linear spacing, keeping grid search tractable.

***Comparison with PTQD (mdUQ)***: The authors provided a detailed explanation (lines 209–216) distinguishing Q-Sched from PTQD, including differences in assumptions (Gaussian-free), objectives (final-image vs. intermediate step), and correction targets.

**Outstanding Concerns**\
Despite a strong rebuttal, several substantive concerns remain:

***Limited conceptual novelty***: While reviewers appreciated the idea of scheduler-level PTQ, Q-Sched’s core mechanism—tuning scalar coefficients via grid search—is relatively simple and resembles well-known bias-variance scaling ideas. The JAQ loss, while effective, is an empirical combination of metrics and lacks a deeper theoretical grounding.

***Marginal incremental gains over strong baselines***: Despite iso-latency, Q-Sched’s performance gains are modest—often in the range of 0.5–1.5% on FID or CLIPScore—raising questions about whether the method justifies a full paper in terms of impact.

***Calibration-set dependency***: The method still requires a calibration set, and while the authors try to mitigate overfitting via sDCI prompts, this remains a limitation for deployment in unseen or domain-shifted settings.

***Lack of end-to-end quantization***: The method applies only to the scheduler, not the model weights, and does not offer a holistic solution for quantizing diffusion models. Other methods (e.g., PTQD, MixDQ) offer more comprehensive adaptation strategies.

**Reviewer Scores:**

**Reviewer 53Sq (Initial Score: 4)**: Acknowledged novelty but raised concerns about latency and comparison to small models. Despite rebuttal, final comments suggest persisting doubts about real-world applicability. Likely Final Score: 4 (unchanged)

**Reviewer pBRf (Initial Score: 6)**: Generally positive and appreciated the simplicity and effectiveness of Q-Sched. Concerns about JAQ loss were addressed reasonably. Likely Final Score: 6 (unchanged)

**Reviewer mdUQ (Initial Score: 4)**: Score was raised during rebuttal. However, initial skepticism around novelty and scalability suggests the final score is still borderline. Final Score: 6

**Reviewer aN57 (Initial Score: 6)**: Positively inclined but raised concerns about simplicity, expressivity, and overfitting, which were partially addressed. Likely Final Score: 6 (unchanged)

---

### Decision · Program_Chairs · 2026-01-26

Reject